# The Underappreciated Power of Vision Models for Graph Structural Understanding

**Xinjian Zhao**[§ ¶*], **Wei Pang**[§ *], **Zhongkai Xue**[§*], **Xiangru Jian**[◇ *],
**Lei Zhang**[§], **Yaoyao Xu**[§], **Xiaozhuang Song**[§], **Shu Wu**[¶†]**, Tianshu Yu**[§†]

[§] School of Data Science, The Chinese University of Hong Kong, Shenzhen
[¶] Institute of Automation, Chinese Academy of Sciences
[◇] Cheriton School of Computer Science, University of Waterloo
`{xinjianzhao1,weipang,zhongkaixue}@link.cuhk.edu.cn, xiangru.jian@uwaterloo.ca`
`{leizhang1,yaoyaoxu,xiaozhuangsong1}@link.cuhk.edu.cn`
`shu.wu@nlpr.ia.ac.cn, yutianshu@cuhk.edu.cn`

## Abstract

Graph Neural Networks operate through bottom-up message-passing, fundamentally differing from human visual perception, which intuitively captures global structures first. We investigate the underappreciated potential of vision models for graph understanding, finding they achieve performance comparable to GNNs on established benchmarks while exhibiting distinctly different learning patterns. These divergent behaviors, combined with limitations of existing benchmarks that conflate domain features with topological understanding, motivate our introduction of **GraphAbstract**. This benchmark evaluates models' ability to perceive global graph properties as humans do: recognizing organizational archetypes, detecting symmetry, sensing connectivity strength, and identifying critical elements. Our results reveal that vision models significantly outperform GNNs on tasks requiring holistic structural understanding and maintain generalizability across varying graph scales, while GNNs struggle with global pattern abstraction and degrade with increasing graph size. This work demonstrates that vision models possess remarkable yet underutilized capabilities for graph structural understanding, particularly for problems requiring global topological awareness and scale-invariant reasoning. These findings open new avenues to leverage this underappreciated potential for developing more effective graph foundation models for tasks dominated by holistic pattern recognition. The code is available at
`https://github.com/LOGO-CUHKSZ/GraphAbstract`

## 1 Introduction

Graphs are powerful abstractions to represent complex relationships across entities like social networks [62] and molecular structures [24, 30]. Learning effective representations of these graphs is crucial for node classification, graph classification, and link prediction tasks [51, 67, 42, 34, 54]. Over the past decade, Graph Neural Networks (GNNs) have become the dominant paradigm for graph representation learning, demonstrating impressive performance through their message-passing mechanism that iteratively aggregates local neighborhood information [37, 24, 27, 59, 71]. Despite its success, message passing faces several key limitations, such as limited expressiveness [71, 52, 78] and difficulty in capturing long-range dependencies [2]. Numerous efforts have been trying to address different aspects of the message passing mechanism limitations, graph transformer architectures have

---

[*] Equal Contribution
[†] Corresponding authors

39th Conference on Neural Information Processing Systems (NeurIPS 2025).

enriched this paradigm by incorporating long-range interactions across broader graph contexts [16, 39, 74, 68, 50], positional encodings [76, 17, 31] to inject structural priors to improve global topological awareness and graph rewiring techniques mitigate structural bottlenecks in graphs by modifying the topology to enhance information diffusion [58, 48, 5].

However, these advances remain fundamentally constrained by a cognitive discrepancy: While humans intuitively perceive global structures through visual Gestalt principles - understanding the whole before analyzing individual parts [57], GNNs and their variants operate through bottom-up processing. Humans instantly recognize ring structures, symmetric structures, and critical bridges in graph visualizations, yet even advanced architectures struggle with such basic topological understanding [29, 79]. Therefore, although the existing graph learning architecture innovations have made significant progress from both theoretical and practical perspectives, the essence of these efforts is still to identify and repair different aspects of the inherent defects in the message passing mechanism, with limited exploration of vision-based approaches for graph learning despite their natural alignment with how humans perceive network structures.

The flourishing field of computer vision has produced a rich ecosystem of powerful models and techniques that excel at holistic pattern recognition and global structure understanding [45, 7, 36, 38]. This vibrant research landscape provides fertile ground for re-imagining graph learning through a visual lens. Inspired by this potential, we explore a fundamentally different approach: leveraging powerful vision models by translating graph topology into the visual domain. By simply rendering graphs as images using standard layout algorithms [22, 14], we apply vision encoders to graph-level tasks without any graph-specific architectural modifications. This vision-based approach mirrors how humans perceive graph structures visually rather than through explicit message-passing operations.

Our evaluations demonstrate that pure vision encoders perform comparably to specialized GNNs on established graph benchmarks, despite having no graph-specific inductive biases or architectural design. This finding is remarkable considering that these vision models operate solely on graph layouts without access to node features or explicit connectivity information. However, these traditional benchmarks tightly couple domain knowledge with topology, making it difficult to quantitatively study their separate impacts. As shown by [4, 65], models using fixed-structure expander graphs rather than the true molecular topology can match or exceed performance on multiple molecular benchmarks, suggesting that node features often dominate over structural information in standard datasets, potentially masking differences in how models perceive graph topology.

To rigorously evaluate topological understanding, we introduce **GraphAbstract**, a benchmark designed to evaluate how well models perceive graph structures in ways that mirror human visual cognition. We present four carefully crafted tasks that challenge models to abstract critical global graph properties in ways humans naturally do: recognizing organizational archetypes, detecting symmetry patterns, sensing connectivity strength, and identifying critical structural elements. For each task, we meticulously design diverse graph families with well-controlled topological properties. Our evaluation protocol directly tests out-of-distribution (OOD) generalization by systematically increasing graph sizes from training to testing, evaluating whether models can recognize the same structural patterns regardless of scale, a fundamental capability of human cognition.

Our experiments yield several key findings: On tasks requiring the abstraction of global graph properties, vision models demonstrate significant advantages and superior generalization capabilities across different graph scales compared to GNNs. For GNNs, we find that positional encoding methods that inject structural priors achieve substantial improvements over architectural innovations in message passing. These complementary findings point to a unified insight: successful graph understanding stems from accessing global topological information, either through structural priors or visual perception. This "global-first" approach aligns more closely with human cognitive processes for graph understanding, suggesting that future advances in graph representation learning and foundation models may benefit from prioritizing global structural perception over refining local message-passing mechanisms. These results open new perspectives for developing graph learning systems that are more effective, interpretable, and capable of generalizing to complex topological structures.

Our main contributions include: **(1)** demonstrating that vision-based models, which better align with human cognition, achieve strong performance without any graph-specific architectural modifications; **(2)** introducing a rigorously designed benchmark with diverse graph families that evaluates models' ability to understand graph structures and generalize across varying scales, which is a fundamental capability of human cognition; and **(3)** revealing that "global-first" approaches significantly outperform

traditional message-passing innovations, suggesting promising directions for visual-centric graph understanding.

## 2    Related Work

**Graph visualization and graph learning**. Graph visualization and graph learning have traditionally advanced along separate paths. Visualization focuses on creating human-interpretable spatial representations that produce graph layouts through techniques like spectral methods and force-directed algorithms, while graph learning focuses on feature extraction with GNNs. Recent studies indicate a convergence between these domains [63, 82, 13, 40, 64]. For example, GITA [63], GraphTMI [13], and DPR [40] incorporate visual graph layouts into vision-language models to improve graph reasoning tasks, DEL [82] introduces probabilistic layout sampling to enhance GNN expressivity, and domain-specific approaches explore visual representations for molecular graphs [77, 69]. However, these works directly introduced graph layout into existing architectures such as VLMs and GNNs, treating graph layouts and vision models as black boxes without exploring their fundamental cognitive differences in graph understanding.

**Graph learning benchmarks.** Graph representation learning has evolved through two complementary benchmark traditions. The first utilizes real-world datasets from citation networks [73], molecules [47], and comprehensive collections like OGB [30] to evaluate practical performance. The second employs synthetic expressivity tests to probe theoretical limitations through challenges like graph isomorphism [1, 3, 61], substructure counting [9]. Recent LLM-oriented graph benchmarks focus on structural analysis tasks such as node counting, cycle detection, and path finding [60, 6, 44]. However, these benchmarks typically rely on a limited set of random graph generators (e.g., Erdős–Rényi, Watts-Strogatz) that inadequately capture topological diversity [60, 40, 19, 44, 8, 70]. The critical gap in existing benchmarks is their inability to assess human-like graph cognition, specifically, the intuitive ability to recognize and abstract connectivity patterns across varying scales and contexts.

For brevity, we provide extended discussions of GNN architectures and graph positional encoding methods in Appendix B.

## 3    Visual vs. Message-Passing: Distinct Cognitive Patterns

We analyze the differences between two model families (GNNs and vision models) in graph tasks from three perspectives: prediction overlap, interpretability case study, and training dynamics & prediction confidence. All models are trained on five graph classification benchmarks [47], with implementation details provided in Appendix J.1.

**Prediction Overlap Analysis.** Our experiments across five datasets show that vision models achieve performance competitive with GNNs, as shown in Table 1. However, Figure 1 reveals that while GNN variants behave similarly to each other, GNNs and vision models show distinct prediction patterns: they not only differ in their correct predictions, but often succeed on samples where the other fails. This consistent pattern across all datasets suggests that these two model families develop fundamentally different strategies for processing graph information. The stark contrast in their prediction patterns indicates that GNNs and vision models might be capturing different aspects of graph structure, motivating a deeper investigation into their respective strengths and limitations.

**Case Studies**. We conduct case studies across multiple benchmark datasets, with detailed analysis presented here using the first three samples from the test set of PROTEINS dataset in Figure 3. We leverage GNN Explainer [75] for graph models and Grad-CAM [53] for vision models, generating visualizations across multiple network layers (2/3/4-layer for GNNs, and low/mid/high-level features for vision models). The three cases represent distinct structural scenarios commonly found in protein graphs: hierarchical local-to-global patterns, critical bridge structures, and chain-like configurations. Through these visualizations, we observe vision models' remarkable adaptability: exhibiting progressive focus from local to regional features in hierarchical structures, maintaining consistent attention on critical bridges across all levels, and adopting a global-centric strategy for chain-like patterns. This flexible processing strategy stands in sharp contrast to GNNs, where GNN Explainer shows relatively uniform attention patterns constrained by local message passing. Notably, our studies on ENZYMES dataset (see Appendix J.3) reveal that vision models' attention regions align better with previously identified discriminative patterns [12] compared to GNN-based approaches.

Table 1: Performance comparison on different datasets. Results show the accuracy (%) of different models, reported as mean $\pm$ std over 5 runs.

| Model | NCI1 | IMDB-MULTI | ENZYMES | IMDB-BINARY | PROTEINS |
|---|---|---|---|---|---|
| **GNN Models** | | | | | |
| GAT | $67.0 \pm 0.5$ | $49.7 \pm 1.0$ | $27.0 \pm 5.9$ | $69.4 \pm 2.2$ | $74.1 \pm 1.5$ |
| GIN | $70.4 \pm 1.3$ | $50.9 \pm 1.7$ | $26.3 \pm 3.9$ | $70.0 \pm 3.0$ | $75.0 \pm 1.1$ |
| GCN | $65.4 \pm 1.1$ | $48.4 \pm 1.8$ | $25.3 \pm 2.2$ | $71.0 \pm 1.3$ | $74.8 \pm 1.1$ |
| GPS | $\mathbf{76.3 \pm 3.4}$ | $50.1 \pm 2.2$ | $34.3 \pm 5.6$ | $69.8 \pm 2.1$ | $76.0 \pm 1.2$ |
| **Vision Models** | | | | | |
| RESNET | $67.7 \pm 1.0$ | $51.2 \pm 1.1$ | $36.3 \pm 1.6$ | $72.2 \pm 2.2$ | $79.6 \pm 2.5$ |
| VIT | $63.5 \pm 1.4$ | $50.8 \pm 3.3$ | $27.3 \pm 3.9$ | $71.8 \pm 2.8$ | $\mathbf{83.1 \pm 0.9}$ |
| SWIN | $69.0 \pm 0.5$ | $52.0 \pm 2.0$ | $40.3 \pm 3.9$ | $68.8 \pm 3.5$ | $81.8 \pm 1.8$ |
| ConvNeXt | $70.2 \pm 2.2$ | $\mathbf{53.5 \pm 1.3}$ | $\mathbf{41.7 \pm 3.2}$ | $\mathbf{73.8 \pm 1.6}$ | $80.7 \pm 2.6$ |

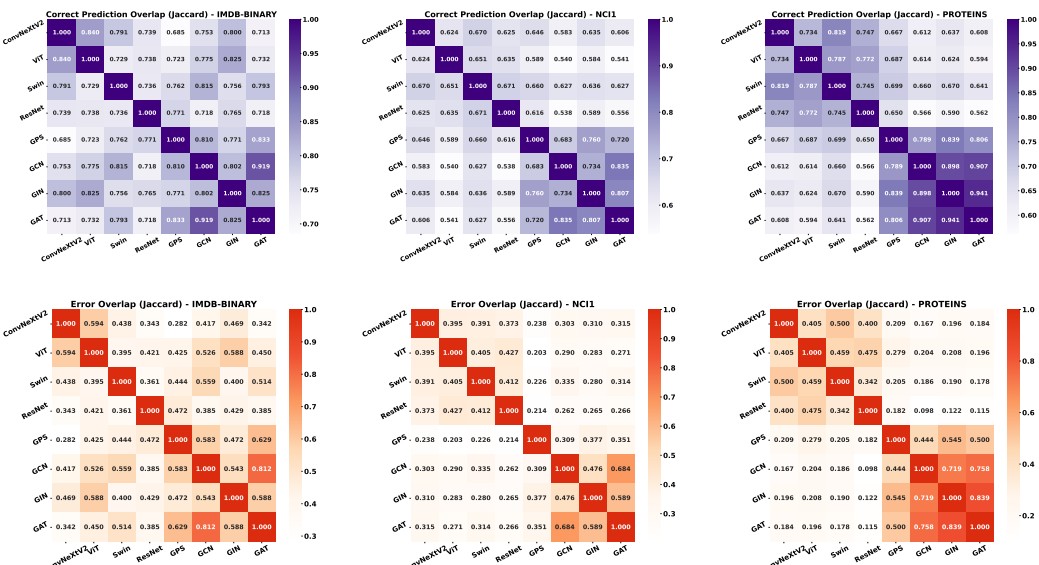

Figure 1: Prediction overlap analysis across different datasets. Top row: correct prediction overlap; Bottom row: error pattern overlap. GNN variants show high internal consistency, suggesting homogeneous learning behavior, while GNN and Vision models exhibit distinct prediction patterns.

**Training dynamics & confidence**. We report the training curves in Figure 3 and provide detailed results for all datasets in Appendix J.2. The training dynamics reveal striking differences in learning behaviors across architectures. Vision models, regardless of their specific architectures, demonstrate strong memorization capabilities but with different learning speeds. CNN-based models show aggressive training dynamics, rapidly achieving near-perfect training accuracy while their training loss approaches zero. Transformer-based models exhibit somewhat slower learning progression, but ultimately reach similar training performance levels. However, all vision models suffer from substantial generalization gaps, with validation accuracy significantly lower than their training performance. GNN variants display notably different learning patterns. Traditional GNNs show more modest training performance, with both training and validation metrics evolving gradually and plateauing at lower levels. The GPS model, featuring additional global processing capabilities, achieves higher training accuracy but still exhibits limited validation performance similar to other GNN variants. These empirical observations suggest that the key challenge in vision-based graph learning lies not in improving models' pattern recognition capabilities, which are already remarkably strong, but in developing effective mechanisms to bridge the substantial memorization-generalization gap. This consistent pattern across all vision architectures points to a fundamental challenge that may require solutions beyond architectural modifications alone, such as specialized pre-training strategies or graph-specific data augmentation techniques.

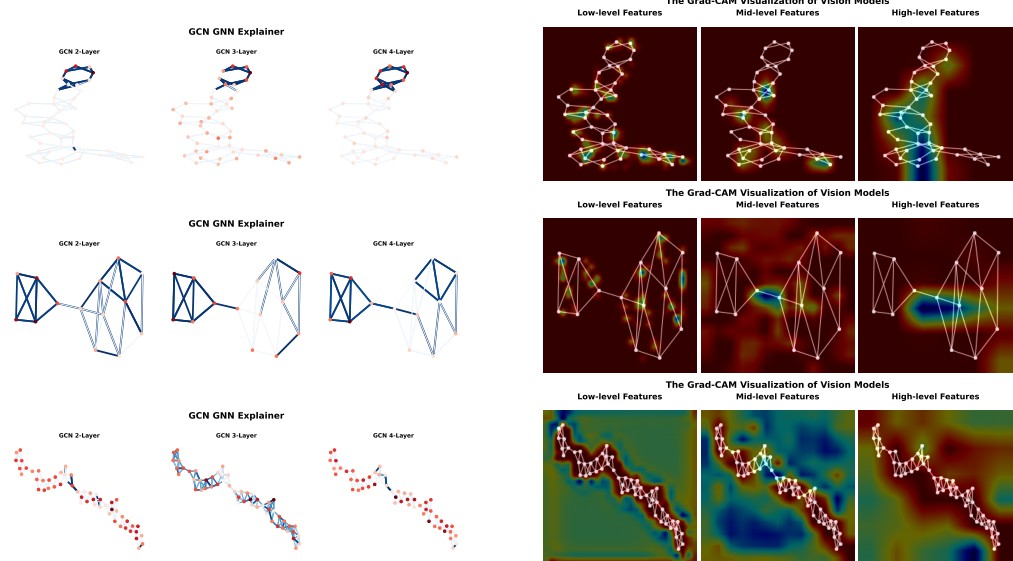

Figure 2: Case Studies for PROTEINS dataset.

**Why an intuitive topology benchmark is essential.** Our analysis shows that different types of models process graphs differently, but existing benchmarks fail to measure how closely these approaches align with human visual perception of graph structures. Current evaluations mix structural understanding with domain-specific features, leading to cases where models perform well even with random graph topologies [4]. This gap between intuitive human understanding and current evaluation practices highlights the need for a dedicated benchmark to specifically assess intuitive topological perception.

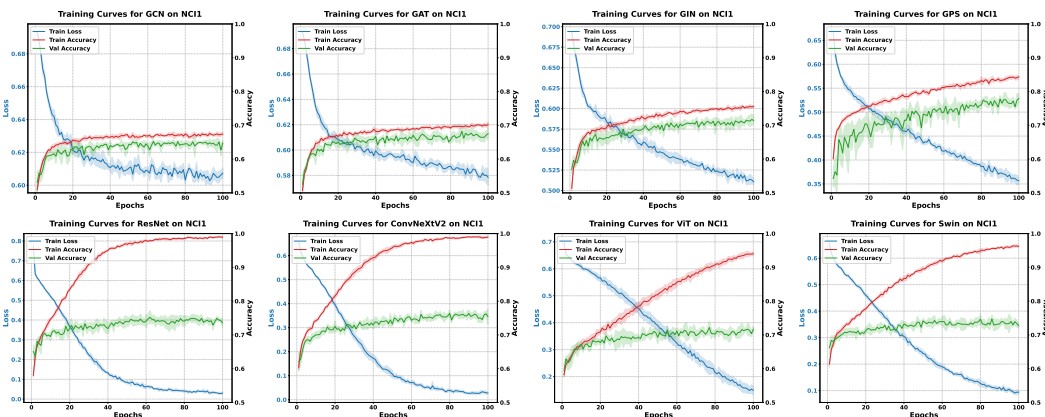

Figure 3: Training dynamics across different architectures on NCI1 dataset. For each model, we plot the training loss (blue), training accuracy (red), and validation accuracy (green) over 100 epochs. The shaded areas represent the standard deviation across multiple runs.

## 4 A New Benchmark: GraphAbstract

### 4.1 Motivation and Design Principles

Building on our analysis of cognitive divergence between model families, we introduce **GraphAbstract**, a benchmark specifically designed to evaluate fundamental graph understanding capabilities that align with human visual reasoning. Traditional benchmarks in domains like molecular prediction, citation networks, and protein interaction graphs inadvertently couple domain-specific node features with topology, often allowing models to succeed through feature-based shortcuts rather than genuine

structural understanding. This limitation becomes particularly evident in recent studies [4, 65], where models using fixed-structure expander graphs rather than true molecular topologies match or exceed performance on multiple benchmarks. The theoretical evaluation of graph neural networks has primarily centered on the Weisfeiler-Lehman (WL) test, which measures a model's ability to distinguish between pairs of non-isomorphic graphs of the same size. While valuable for theoretical analysis, this approach is inherently limited to binary discrimination between fixed-size graph pairs rather than evaluating models' ability to recognize abstract structural patterns across varying scales and contexts. What remains missing is a rigorous evaluation framework that targets explicitly models' ability to perceive, abstract, and reason about fundamental graph properties in ways that mirror human visual cognition.

**GraphAbstract** addresses these limitations through four carefully designed tasks that isolate pure structural comprehension from domain-specific attributes. Each task evaluates a different aspect of how humans intuitively perceive graphs: recognizing organizational archetypes, detecting symmetry patterns, sensing connectivity strength, and identifying critical structural elements. By focusing on these fundamental perceptual capabilities and systematically varying graph scale between training and testing, we can rigorously assess whether graph learning approaches develop the scale-invariant structural understanding that characterizes human visual cognition.

### 4.2 Benchmark Details

#### 4.2.1 High-level Topology Classification

The first and core task of our benchmark is high-level topology classification, where models must identify the dominant topological pattern in a graph $\mathcal{G} = (V, E)$. We carefully design six fundamental topological patterns commonly observed in real-world networks, each representing distinct organizational principles that humans can readily identify through visual inspection. This task evaluates a model's ability to perceive global structural organization beyond local connectivity patterns, mirroring how humans naturally identify network archetypes across diverse domains.

The following six graph types represent our core taxonomy of high-level topological patterns: The **Cyclic Structure** is generated as an annular random geometric graph where nodes are distributed within a ring-shaped region and connections are established based on spatial proximity [23]. The **Random Geometric Graph** structure emerges from spatial constraints, where connections are determined by the proximity of nodes in an underlying metric space. This topology is ubiquitous in wireless sensor networks, urban infrastructure, and physical systems governed by geographical limitations [49]. **Hierarchical Structures** organize nodes into multiple levels where higher tiers have fewer, more densely connected nodes controlling numerous nodes in lower layers. **Community Structures** feature multiple densely connected subgroups with relatively sparse inter-group connections, representing common patterns in social networks and biological networks [25]. The **Bottleneck Configuration** contains critical narrow passages between larger substructures, similar to traffic networks, information flow channels, and metabolic networks. These structures are particularly important for testing models' ability to identify crucial connecting components that often represent vulnerability points in real systems. **Multicore-periphery Networks** [72] exhibit multiple densely connected centers with their respective peripheral nodes, reflecting patterns found in distributed computing systems, multi-center urban structures, and neural networks.

#### 4.2.2 Symmetry Classification

Symmetry perception represents one of the most fundamental pattern recognition capabilities in human cognition. Humans can readily identify symmetric patterns through visual inspection, even without explicit mathematical analysis, when examining graph structures. This capacity has profound practical importance across domains: in chemical structures, symmetry determines molecular properties and reactivity; in network design, engineers leverage symmetry for resilience and load balancing; while cryptographers specifically construct asymmetric structures to enhance security. Our Symmetry Classification task challenges models to develop this same intuitive capability by determining whether a graph possesses non-trivial symmetry based on its automorphism properties.

To precisely characterize graph symmetry, we employ the concept of graph automorphism. Using the automorphism group, we can precisely categorize graphs based on their symmetry properties:

**Definition 1** (Graph Automorphism). *Given a graph $\mathcal{G} = (V, E)$, an automorphism is a bijection $\phi : V \to V$ such that $(u, v) \in E$ if and only if $(\phi(u), \phi(v)) \in E$. The set of all automorphisms forms a group $\mathrm{Aut}(\mathcal{G})$ under composition:*

$$\mathrm{Aut}(\mathcal{G}) = \{\phi : V \to V \mid \phi \text{ is bijective and } (u, v) \in E \iff (\phi(u), \phi(v)) \in E\} \quad (1)$$

**Definition 2** (Symmetric and Asymmetric Graphs). *A graph is classified as symmetric if $|\mathrm{Aut}(\mathcal{G})| > 1$, indicating the existence of at least one non-identity automorphism, and asymmetric if $|\mathrm{Aut}(\mathcal{G})| = 1$, where the only automorphism is the identity mapping.*

To construct a diverse symmetry classification dataset, we implement several carefully designed generation strategies for both symmetric and asymmetric graphs. As part of our approach, we also extract a collection of base graphs from real-world datasets using multiple sampling strategies to enhance structural diversity, as detailed in Appendix D.1.2. For symmetric graphs, we employ four principled approaches based on group-theoretic constructions. Our first symmetric graph generation method utilizes Cayley graphs, which are constructed from algebraic groups and naturally exhibit rich symmetry properties:

**Definition 3** (Cayley Graph). *Given a group $\Gamma$ and a generating set $S \subset \Gamma$ where $S = S^{-1}$ (closed under inverses), the Cayley graph $\mathrm{Cay}(\Gamma, S)$ has vertices $V = \Gamma$ and edges $E = \{(g, gs) \mid g \in \Gamma, s \in S\}$.*

Using this definition, we construct Cayley graphs $\mathrm{Cay}(\mathbb{Z}_n, S)$ where $\mathbb{Z}_n$ is the cyclic group of order $n$ and $S$ contains generators of the group (elements coprime to $n$). These graphs inherently possess rich symmetry patterns with automorphism groups containing at least $n$ elements.

The second approach leverages bipartite double covers, which provide a systematic way to construct symmetric graphs from arbitrary base graphs:

**Definition 4** (Bipartite Double Cover). *For a graph $\mathcal{G} = (V, E)$, its bipartite double cover $\tilde{\mathcal{G}} = (\tilde{V}, \tilde{E})$ is defined as:*

$$\tilde{V} = V \times \{0, 1\} = \{(v, i) \mid v \in V, i \in \{0, 1\}\} \quad (2)$$

$$\tilde{E} = \{((u, 0), (v, 1)), ((u, 1), (v, 0)) \mid (u, v) \in E\} \quad (3)$$

We generate bipartite double covers from various base graphs, including random graphs, community graphs, bottleneck graphs, and real-world data. Each cover naturally possesses a non-trivial automorphism $\sigma((v, i)) = (v, 1 - i)$ that swaps the two layers, guaranteeing $|\mathrm{Aut}(\tilde{\mathcal{G}})| \geq 2$ and ensuring the graph is symmetric by definition. Appendix E provides a detailed proof of this property.

Our third method creates symmetric structures through the Cartesian product:

**Definition 5** (Cartesian Product). *For graphs $\mathcal{G}_1 = (V_1, E_1)$ and $\mathcal{G}_2 = (V_2, E_2)$, their Cartesian product $\mathcal{G}_1 \square \mathcal{G}_2 = (V, E)$ is defined as:*

$$V = V_1 \times V_2 = \{(u, v) \mid u \in V_1, v \in V_2\} \quad (4)$$

$$E = \{((u_1, v), (u_2, v)) \mid (u_1, u_2) \in E_1, v \in V_2\} \cup \{((u, v_1), (u, v_2)) \mid u \in V_1, (v_1, v_2) \in E_2\} \quad (5)$$

We create symmetric graphs through Cartesian products of known symmetric components (e.g., cycle graphs $C_n$, path graphs $P_n$, star graphs $S_n$), resulting in structures like prism graphs $C_n \square K_2$ and torus grids $C_m \square C_n$. For products involving real-world graphs (which we also use to generate asymmetric graphs), we rigorously verify and filter based on the actual automorphism group properties, as the Cartesian product preserves symmetry only when both factor graphs are symmetric.

Additionally, we employ multi-layer cyclic covers using real-world data as base graphs. These covers possess a natural cyclic symmetry where the automorphism $\tau((v, i)) = (v, (i + 1) \mod k)$ generates a cyclic group isomorphic to $\mathbb{Z}_k$, ensuring $|\mathrm{Aut}(\mathcal{G}_k)| \geq k$. The full mathematical definition and additional properties of these structures are provided in Appendix E.

For asymmetric graphs, we employ two primary strategies. First, we create perturbed graphs using Double-Edge Swap perturbations [46], where we start with symmetric structures and systematically transform them through edge swaps. Specifically, we repeatedly select two edges $(v_i, v_j)$ and $(v_k, v_l)$ and replace them with $(v_i, v_l)$ and $(v_k, v_j)$ if these new edges don't already exist. After each swap, we

verify both connectivity preservation and symmetry breaking using automorphism group computation. This method maintains the degree distribution of the original graph while disrupting its symmetry structure. Second, we leverage real-world graph patterns through Cartesian products of real-world graphs, whose inherent irregularity typically leads to asymmetric structures.

### 4.2.3 Spectral Gap Regression

While humans cannot directly "see" mathematical properties of networks, we intuitively perceive network conductance, bottleneck structures, and overall connectivity strength through visual inspection. These perceptual judgments closely align with what graph theory formalizes as the spectral gap $\lambda_2(\mathcal{G})$, the second-smallest eigenvalue of the normalized Laplacian matrix [10]. This fundamental parameter quantifies a graph's global connectivity characteristics, determining how quickly random walks mix ($t_{\text{mix}} \propto 1/\lambda_2$) and providing lower bounds on critical connectivity measures. Our regression task challenges models to develop representations that can infer this abstract property directly from topology, mirroring human ability to estimate network efficiency without explicit computation.

To ensure diverse spectral properties, we generate graphs using stochastic block models with varying mixing parameters, geometric graphs with different connection radii, and configuration models with targeted degree distributions. This design forces models to develop structural intuitions equivalent to understanding that bottlenecked networks (low $\lambda_2$) exhibit restricted information flow, while expander-like graphs (high $\lambda_2$) enable rapid diffusion. This reflects precisely the type of reasoning humans employ when analyzing network resilience in domains ranging from transportation systems to communication infrastructure.

### 4.2.4 Bridge Counting

Bridge counting evaluates a model's ability to identify critical edges whose removal would increase the number of connected components in a graph. Formally, given a graph $\mathcal{G} = (V, E)$, we define the set of bridges $\mathcal{B}(\mathcal{G})$ as:

$$\mathcal{B}(\mathcal{G}) = \{e \in E \mid \kappa(\mathcal{G} \setminus \{e\}) > \kappa(\mathcal{G})\} \tag{6}$$

where $\kappa(\mathcal{G})$ denotes the number of connected components in graph $\mathcal{G}$. The objective is to predict $|\mathcal{B}(\mathcal{G})|$, the total number of bridges in the input graph.

This regression task requires models to understand both local edge importance and global connectivity patterns. Bridges serve as critical connectors between biconnected components of a graph, with each bridge $e = (u, v)$ satisfying the property that there exists no alternative path between $u$ and $v$ when $e$ is removed.

The bridge identification challenge varies systematically with graph structure, requiring models to adapt their reasoning across different topological contexts. Models lacking this capability face limitations in practical applications requiring critical connectivity awareness, such as molecular stability analysis and retrosynthetic planning [11, 56].

These four tasks systematically probe models' ability to perceive and reason about fundamental graph properties. While representing a subset of human topological capabilities, they provide diagnostic tests for structural understanding that underlies many practical applications. Systematically measuring these capability gaps offers insights into current limitations and directions for developing more robust graph learning models.

### 4.3 Evaluation Protocol

To rigorously assess the generalization capabilities of graph learning models, we introduce a systematic evaluation framework incorporating progressively challenging distribution shifts based on graph scale. This framework enables us to quantify how well models can transfer topological understanding across varying graph sizes, a capability that humans demonstrate naturally.

Our evaluation includes three test settings of increasing difficulty: **ID** (In-Distribution) setting uses test graphs containing 20-50 nodes, matching the training distribution. **Near-OOD** (Near Out-of-Distribution) setting contains graphs with 40-100 nodes, representing a moderate scale shift. **Far-OOD** (Far Out-of-Distribution) setting features graphs with 60-150 nodes, constituting a significant scale shift. These examples challenge models to recognize the same topological patterns at dramatically larger scales, testing their ability to abstract core structural principles independent of

Table 2: Performance comparison across different tasks and models. First and second best performances are highlighted in each setting and model family.

| Model | Topology | | | Symmetry | | | Spectral | | | Bridge | | |
|---|---|---|---|---|---|---|---|---|---|---|---|---|
| | ID | Near-OOD | Far-OOD | ID | Near-OOD | Far-OOD | ID | Near-OOD | Far-OOD | ID | Near-OOD | Far-OOD |
| GCN+Degree | 80.67 ± 0.60 | 54.67 ± 2.69 | 33.67 ± 3.56 | 69.73 ± 0.87 | 66.87 ± 1.08 | 65.13 ± 1.94 | 0.1325 ± 0.0022 | 0.2517 ± 0.0039 | 0.3167 ± 0.0046 | 1.3995 ± 0.0294 | 3.1067 ± 0.1171 | 5.6302 ± 0.0898 |
| GPS+Degree | 81.40 ± 1.81 | 64.33 ± 2.18 | 37.87 ± 4.60 | 72.73 ± 0.87 | 70.87 ± 1.52 | 66.53 ± 2.44 | 0.0696 ± 0.0020 | 0.1844 ± 0.0136 | 0.4271 ± 0.0494 | 1.5226 ± 0.1512 | 3.3043 ± 0.2873 | 5.9010 ± 0.3702 |
| GIN+Degree | 79.87 ± 1.05 | 62.47 ± 2.79 | 39.33 ± 4.31 | 71.57 ± 1.54 | 69.13 ± 1.54 | 68.47 ± 0.51 | 0.1159 ± 0.0025 | 0.2885 ± 0.0474 | 0.6460 ± 0.2373 | 1.2953 ± 0.0373 | 3.3695 ± 0.3631 | 7.0563 ± 1.0770 |
| GAT+Degree | 81.87 ± 1.67 | 58.40 ± 3.52 | 42.80 ± 3.07 | 69.47 ± 0.89 | 67.40 ± 0.76 | 65.70 ± 1.36 | 0.1329 ± 0.0004 | 0.2512 ± 0.0133 | 0.3149 ± 0.0158 | 1.3775 ± 0.0472 | 3.1871 ± 0.1867 | 5.6034 ± 0.1339 |
| GCN+LapPE | 86.83 ± 1.64 | 70.97 ± 3.99 | 55.00 ± 3.08 | 68.63 ± 1.02 | 66.19 ± 0.88 | 65.35 ± 2.30 | 0.0442 ± 0.0211 | 0.1076 ± 0.0526 | 0.1840 ± 0.0807 | 0.9961 ± 0.0762 | 2.8534 ± 0.2058 | 5.7156 ± 0.3173 |
| GPS+LapPE | 93.07 ± 1.14 | 81.00 ± 2.77 | 47.40 ± 2.16 | 71.52 ± 1.50 | 69.89 ± 1.58 | 66.80 ± 1.88 | 0.0263 ± 0.0084 | 0.0706 ± 0.0313 | 0.1781 ± 0.0881 | 1.1665 ± 0.6545 | 2.4846 ± 0.5410 | 5.3825 ± 0.6990 |
| GIN+LapPE | 93.37 ± 0.62 | 82.13 ± 2.96 | 51.13 ± 4.34 | 71.37 ± 1.19 | 68.83 ± 1.17 | 67.17 ± 1.55 | 0.0217 ± 0.0057 | 0.0538 ± 0.0145 | 0.1268 ± 0.0415 | 0.8683 ± 0.1112 | 2.4427 ± 0.2064 | 5.1461 ± 0.3576 |
| GAT+LapPE | 84.90 ± 2.77 | 72.07 ± 3.13 | 54.07 ± 7.26 | 69.15 ± 1.09 | 66.46 ± 1.28 | 66.22 ± 1.71 | 0.0182 ± 0.0026 | 0.0419 ± 0.0039 | 0.0722 ± 0.0048 | 0.9603 ± 0.0701 | 2.4669 ± 0.1743 | 5.2778 ± 0.2455 |
| GCN+SignNet | 94.47 ± 1.54 | 94.20 ± 1.65 | 77.93 ± 2.77 | 69.03 ± 0.93 | 67.07 ± 0.83 | 65.60 ± 1.43 | 0.0203 ± 0.0018 | 0.0274 ± 0.0034 | 0.0523 ± 0.0147 | 0.6750 ± 0.1104 | 2.4387 ± 0.5131 | 6.3090 ± 1.5907 |
| GPS+SignNet | 81.80 ± 14.38 | 87.67 ± 5.19 | 75.20 ± 6.53 | 70.73 ± 1.46 | 69.47 ± 1.24 | 67.73 ± 1.03 | 0.0244 ± 0.0060 | 0.0783 ± 0.0134 | 0.3133 ± 0.0529 | 0.9872 ± 0.3033 | 1.9819 ± 0.4649 | 4.5278 ± 0.8672 |
| GIN+SignNet | 94.20 ± 1.80 | 84.73 ± 6.50 | 61.00 ± 8.72 | 70.43 ± 0.69 | 68.60 ± 2.14 | 67.50 ± 2.11 | 0.0237 ± 0.0039 | 0.0750 ± 0.0195 | 0.2417 ± 0.0904 | 0.6303 ± 0.1828 | 2.4745 ± 0.3013 | 7.1992 ± 1.0679 |
| GAT+SignNet | 94.00 ± 1.21 | 96.47 ± 1.60 | 85.27 ± 6.83 | 69.90 ± 0.98 | 67.60 ± 1.47 | 67.27 ± 1.37 | 0.0204 ± 0.0047 | 0.0303 ± 0.0030 | 0.0571 ± 0.0154 | 0.5713 ± 0.0973 | 1.7152 ± 0.1943 | 4.1380 ± 0.2873 |
| GCN+SPE | 93.20 ± 2.16 | 90.60 ± 4.77 | 72.33 ± 7.90 | 68.90 ± 0.79 | 66.80 ± 1.40 | 64.80 ± 2.85 | 0.0255 ± 0.0025 | 0.0507 ± 0.0039 | 0.1351 ± 0.0284 | 0.5503 ± 0.0777 | 2.4143 ± 0.2405 | 3.8632 ± 1.2460 |
| GPS+SPE | 84.80 ± 13.75 | 84.07 ± 16.04 | 72.20 ± 14.51 | 71.97 ± 1.65 | 70.67 ± 1.23 | 67.70 ± 1.37 | 0.0681 ± 0.0298 | 0.1537 ± 0.0839 | 0.6716 ± 0.2709 | 0.6402 ± 0.1753 | 1.4666 ± 0.0713 | 3.8021 ± 1.0492 |
| GIN+SPE | 94.53 ± 1.76 | 87.80 ± 9.89 | 70.33 ± 12.09 | 70.87 ± 1.11 | 68.80 ± 1.12 | 68.63 ± 0.97 | 0.0376 ± 0.0028 | 0.1491 ± 0.0382 | 0.8412 ± 0.3343 | 0.6011 ± 0.1649 | 2.4499 ± 0.6701 | 7.8487 ± 2.0425 |
| GAT+SPE | 93.53 ± 4.66 | 92.60 ± 6.95 | 85.33 ± 9.94 | 68.07 ± 1.07 | 66.87 ± 0.98 | 67.47 ± 0.64 | 0.0296 ± 0.0029 | 0.0784 ± 0.0044 | 0.2210 ± 0.0347 | 0.4854 ± 0.0622 | 1.5176 ± 0.4072 | 4.1430 ± 1.7660 |
| Swin | 94.80 ± 0.54 | 97.73 ± 0.57 | 89.13 ± 3.26 | 92.50 ± 0.43 | 90.77 ± 0.81 | 84.70 ± 1.36 | 0.0312 ± 0.0037 | 0.0594 ± 0.0024 | 0.0946 ± 0.0094 | 0.6526 ± 0.0547 | 1.6338 ± 0.1675 | 3.7918 ± 0.3361 |
| ConvNeXtV2 | 95.20 ± 0.34 | 97.20 ± 1.48 | 90.33 ± 4.60 | 92.83 ± 0.53 | 89.13 ± 0.57 | 84.67 ± 0.77 | 0.0279 ± 0.0047 | 0.0578 ± 0.0056 | 0.1006 ± 0.0047 | 0.6261 ± 0.0702 | 1.8045 ± 0.2007 | 4.1809 ± 0.2742 |
| ResNet | 95.87 ± 0.62 | 96.27 ± 1.02 | 87.44 ± 3.33 | 93.47 ± 0.66 | 88.83 ± 0.64 | 84.20 ± 0.39 | 0.0335 ± 0.0021 | 0.0600 ± 0.0063 | 0.1102 ± 0.0100 | 0.7771 ± 0.1095 | 1.6356 ± 0.1643 | 3.6814 ± 0.1217 |
| ViT | 94.00 ± 0.99 | 95.20 ± 1.20 | 86.40 ± 1.61 | 94.03 ± 1.04 | 91.03 ± 0.56 | 85.67 ± 1.06 | 0.0345 ± 0.0046 | 0.0746 ± 0.0081 | 0.1154 ± 0.0080 | 0.7406 ± 0.1167 | 1.8263 ± 0.0679 | 4.3765 ± 0.1214 |

scale. This evaluation framework serves as an analog to human cognitive flexibility, where people can seamlessly recognize familiar patterns at vastly different scales. For instance, humans can readily identify the same community structure whether it appears in a small departmental network of dozens of people or a large organizational chart with hundreds of employees. The ability to maintain consistent performance across these distribution shifts reflects the kind of scale-invariant understanding that advanced graph reasoning systems should aspire to develop.

**Baselines.** We evaluate two primary model families: graph neural networks and vision-based models. For GNNs, we implement four architectures using one-hot degree encoding as node features: GCN [37], GIN [71], GAT [59], and GPS [50], combined with three positional encoding schemes: LapPE [17], SignNet [41], SPE [31]. For vision-based approaches, we evaluate four backbone architectures: ResNet-50 [28], Swin Transformer-Tiny [43], ViT-B/16 [15], and ConvNeXtV2-Tiny [66], with three graph layout algorithms: Kamada-Kawai [35], Spectral layout [26], and ForceAtlas2 [33]. Benchmark statistics, implementation details, and examples of graphs from all four tasks are provided in Appendices C, D, and H, respectively.

## 4.4 Main Results

Our benchmark isolates the challenge of topology understanding from feature-based learning prevalent in tasks like node classification and molecular property prediction. This design enables us to focus specifically on evaluating models' capability to comprehend graph structural patterns. Our extensive experiments reveal several key findings.

*Vision Models Exhibit Superior Scale-Invariant Understanding.* Tables 2 and 3 demonstrate that pure vision-based models show remarkable proficiency in abstracting global topological patterns across varying scales. Vision models maintain consistent performance across increasing distribution shifts, while GNNs exhibit severe degradation. On topology classification, vision models drop only 5-6% accuracy from ID to Far-OOD settings, while basic GNNs with one-hot degree feature experience dramatic declines of over 45%. This stark contrast highlights vision models' human-like ability to recognize organizational patterns regardless of scale. The vision advantage is particularly pronounced in symmetry detection, where vision models with spectral layouts achieve 20% higher accuracy than even the best GNN variants. This task directly evaluates a model's capacity to perceive global structural properties that humans naturally identify through visual inspection.

*Layout Algorithms Critically Shape Visual Graph Understanding.* Different layout algorithms significantly impact how easily humans and models perceive key structural properties. Spectral layouts, for instance, excel at symmetry detection (8-10% higher accuracy than force-directed approaches). Their use of Laplacian eigenvectors often leads to node overlap when visualizing graphs with symmetries or highly repetitive patterns. This simplification of complex structures into more recognizable forms makes global properties more apparent (detailed in Appendix I.2). Similarly, circular layouts [21], with detailed performance reported in Appendix I.3, arrange nodes in perfect circles. This allows for straightforward symmetry assessment through edge density: symmetric graphs show uniform connections, while in asymmetric graphs, humans can easily spot irregular densities or specific edges that break the symmetry. In contrast, force-directed methods like Kamada Kawai prioritize preserving the graph's intrinsic shape by optimizing for clear visual

Table 3: Performance comparison between KK, ForceAtlas2, and Spectral layout algorithms across different tasks and vision models

| Model | KK | Topology ForceAtlas2 | Spectral | KK | Symmetry ForceAtlas2 | Spectral | KK | Spectral ForceAtlas2 | Spectral | KK | Bridge ForceAtlas2 | Spectral |
|---|---|---|---|---|---|---|---|---|---|---|---|---|
| **ID** | | | | | | | | | | | | |
| Swin | 93.80 ± 1.31 | **94.80 ± 0.54** | 87.27 ± 0.57 | 85.07 ± 1.05 | 80.93 ± 1.09 | **92.50 ± 0.43** | 0.0324 ± 0.0035 | **0.0312 ± 0.0037** | 0.0333 ± 0.0025 | **0.6526 ± 0.0547** | 0.9867 ± 0.1587 | 1.2524 ± 0.0543 |
| ConvNeXtV2 | **95.20 ± 0.34** | 94.93 ± 0.13 | 87.53 ± 0.78 | 87.30 ± 1.20 | 80.73 ± 0.98 | **92.83 ± 0.53** | 0.0284 ± 0.0024 | **0.0279 ± 0.0047** | 0.0403 ± 0.0032 | **0.6261 ± 0.0702** | 0.9226 ± 0.0540 | 1.3579 ± 0.0320 |
| ResNet | **95.87 ± 0.62** | 94.93 ± 1.08 | 85.67 ± 0.47 | 85.63 ± 0.84 | 79.53 ± 0.90 | **93.47 ± 0.66** | **0.0335 ± 0.0021** | 0.0376 ± 0.0098 | 0.0463 ± 0.0071 | **0.7771 ± 0.1095** | 1.0602 ± 0.0759 | 1.5106 ± 0.1855 |
| ViT | **94.00 ± 0.99** | 92.93 ± 0.65 | 86.13 ± 0.65 | 86.47 ± 1.75 | 80.07 ± 1.96 | **94.03 ± 1.04** | 0.0367 ± 0.0045 | **0.0345 ± 0.0046** | 0.0441 ± 0.0067 | **0.7406 ± 0.1167** | 1.0883 ± 0.0887 | 1.3944 ± 0.0493 |
| **Near-OOD** | | | | | | | | | | | | |
| Swin | **97.73 ± 0.57** | 92.20 ± 0.65 | 93.27 ± 1.77 | 81.80 ± 1.31 | 79.87 ± 0.51 | **90.77 ± 0.81** | 0.0819 ± 0.0141 | **0.0594 ± 0.0024** | 0.0690 ± 0.0084 | **1.6338 ± 0.1675** | 2.2916 ± 0.1672 | 2.4495 ± 0.2141 |
| ConvNeXtV2 | **97.20 ± 1.48** | 92.40 ± 0.83 | 93.20 ± 1.89 | 81.83 ± 1.51 | 80.70 ± 0.49 | **89.13 ± 0.57** | 0.0728 ± 0.0122 | **0.0578 ± 0.0056** | 0.0750 ± 0.0109 | **1.8045 ± 0.2007** | 2.4521 ± 0.1678 | 2.5207 ± 0.1870 |
| ResNet | **96.27 ± 1.02** | 93.67 ± 1.15 | 94.60 ± 1.00 | 82.60 ± 1.05 | 79.07 ± 0.67 | **88.83 ± 0.64** | 0.0936 ± 0.0120 | **0.0600 ± 0.0063** | 0.0914 ± 0.0052 | **1.6356 ± 0.1643** | 2.3661 ± 0.2173 | 2.9001 ± 0.3499 |
| ViT | **95.20 ± 1.20** | 92.53 ± 0.88 | 92.67 ± 1.01 | 82.47 ± 1.89 | 79.87 ± 1.31 | **91.03 ± 0.56** | 0.1058 ± 0.0132 | **0.0746 ± 0.0081** | 0.0828 ± 0.0068 | **1.8263 ± 0.0679** | 2.4940 ± 0.1665 | 2.6900 ± 0.0848 |
| **Far-OOD** | | | | | | | | | | | | |
| Swin | **89.13 ± 3.26** | 81.93 ± 1.73 | 87.13 ± 2.95 | 74.20 ± 2.05 | 77.23 ± 1.15 | **84.70 ± 1.36** | 0.1668 ± 0.0142 | 0.1182 ± 0.0050 | **0.0946 ± 0.0094** | **3.7918 ± 0.3361** | 4.9075 ± 0.2818 | 4.9141 ± 0.2744 |
| ConvNeXtV2 | **90.33 ± 4.60** | 87.47 ± 1.39 | 89.00 ± 1.62 | 75.90 ± 2.30 | 77.03 ± 1.10 | **84.67 ± 0.77** | 0.1419 ± 0.0149 | **0.1006 ± 0.0047** | 0.1018 ± 0.0112 | **4.1809 ± 0.2742** | 5.3367 ± 0.2295 | 5.0889 ± 0.1948 |
| ResNet | **87.40 ± 3.30** | 76.93 ± 1.16 | 85.20 ± 2.08 | 74.80 ± 1.36 | 74.93 ± 1.10 | **84.20 ± 0.39** | 0.1739 ± 0.0120 | **0.1102 ± 0.0100** | 0.1179 ± 0.0075 | **3.6814 ± 0.1217** | 4.9441 ± 0.2304 | 5.5314 ± 0.4061 |
| ViT | 79.53 ± 0.69 | **86.40 ± 1.61** | 81.80 ± 0.69 | 76.87 ± 2.27 | 74.17 ± 1.55 | **85.67 ± 1.06** | 0.1837 ± 0.0101 | 0.1471 ± 0.0143 | **0.1154 ± 0.0080** | **4.3765 ± 0.1214** | 5.1766 ± 0.1582 | 5.3629 ± 0.1370 |

separation of nodes and sensible edge routing. This makes individual elements and local relationships more distinguishable, often excelling at revealing community structures and general topological arrangements. These observations highlight the importance of task-aware layout selection in visual graph understanding. Different layout algorithms naturally emphasize different structural properties, suggesting opportunities for developing adaptive visualization strategies that match layout choices to specific reasoning objectives. The intuitions behind how layout algorithms enable vision models to access structural information are discussed in Appendix G.

***Global Structural Priors Help GNNs Bridge the Cognitive Gap.*** Our experiments reveal that incorporating positional encodings (PEs) that inject pre-computed global structural information significantly outperforms innovations in message passing architectures alone. This approach represents an alternative "global-first" strategy within the message-passing framework, providing GNNs with structural context before local propagation begins. All three PE schemes substantially improve performance and generalization capability, with some advanced PE-enhanced GNNs approaching vision model performance on topology tasks. This finding suggests a unified insight: successful graph understanding fundamentally requires access to global topological information, whether through visual perception or explicitly injected structural priors. It also indicates promising directions for future development where vision models could potentially benefit from more explicit structural priors through specialized pre-training or augmentation strategies that emphasize key topological features.

***Computational Trade-offs and Model Capacity.*** Vision models require approximately 10× more computational time than GNN+PE approaches in our standard experimental setting (detailed in Table 5 of Appendix F). To better attribute the performance differences, we conducted parameter scaling experiments with GPS+SPE, expanding it to match or exceed the vision model capacity. Consistent with known challenges in scaling GNNs, performance degraded rather than improved (full results in Table 6 of Appendix F). This confirms that the observed advantages stem from architectural differences, not simply parameter count. The two paradigms understand graph structure through fundamentally different mechanisms: GNNs process topology through abstract message passing and structural priors, while vision models directly recognize patterns in visual representations. The superior out-of-distribution generalization of vision models on global structural understanding tasks aligns with this direct pattern recognition approach. When selecting approaches for practical applications, the computational overhead must be weighed against these advantages in scale generalization and structural reasoning.

## 5    Conclusion

Our work reveals the underappreciated power of vision models for graph structural understanding, demonstrating that vision-based methods better align with human cognitive processes in capturing global topological properties. Through **GraphAbstract**, we systematically quantified these advantages on tasks requiring holistic understanding and scale-invariant reasoning, particularly their ability to maintain consistent performance across varying graph scales. These findings establish visual processing as a complementary pathway to traditional graph learning. While GNNs excel through explicit structural priors and domain-specific inductive biases, vision models offer distinctive strengths in perceiving global patterns through direct pattern recognition. This suggests promising directions for graph foundation models that integrate visual perception with structural reasoning, combining the strengths of both paradigms for more robust and generalizable graph understanding.

# 6  ACKNOWLEDGMENTS

This work is supported in part by the National Natural Science Foundation of China (Grant No. 92470113), and the Guangdong Provincial Key Laboratory of Mathematical Foundations for Artificial Intelligence (2023B1212010001).

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

# Appendix

## Contents

# A    Limitations and Future Work

## A.1    Limitations

**Theoretical Foundations for Layout Algorithms.** While our experiments demonstrate that layout algorithms critically shape performance, we lack complete theoretical characterizations of why certain layouts benefit particular reasoning tasks. The relationship between geometric properties of layouts and their learnability by neural networks remains an open theoretical question. Early work by Eades & Lin [18] connected spring algorithms to geometric automorphisms, but the graph visualization community has since focused primarily on human aesthetics rather than machine learning objectives. Establishing rigorous theoretical frameworks connecting layout properties to learning guarantees represents an essential direction for bridging graph visualization and machine learning research.

**Human Cognition Alignment**: While tasks in GraphAbstract are designed to mirror human cognitive capabilities, we do not directly compare model performance with human behavior on these tasks. Future studies could incorporate human experiments (e.g., eye-tracking studies or timed reasoning tasks) to establish quantitative benchmarks for cognitive alignment.

## A.2    Future Work

**Vision-centric Graph Foundation Models.** Our work establishes visual processing as a viable pathway for graph understanding, demonstrating competitive performance and superior scale generalization on structural reasoning tasks. Realizing this potential requires developing a comprehensive ecosystem for vision-based graph learning, ultimately enabling vision-centric graph foundation models.

Key directions include curating large-scale pretraining datasets of graph visualizations across diverse domains, designing graph-specific augmentation strategies that preserve topological properties in the visual domain, developing visual encoding schemes for node and edge attributes (e.g., color mapping, size encoding, visual markers), creating specialized architectures optimized for processing graph images, and exploring new application scenarios such as interactive graph visualization and visual graph analytics. Such infrastructure could enable foundation models that leverage visual perception for robust and generalizable structural understanding while incorporating rich semantic information.

# B    Extended Related Work

**Graph Neural Networks and Positional Encodings.** Graph Neural Networks have achieved remarkable success across diverse domains through learnable aggregation of neighborhood information [37, 24, 27, 59, 71]. The field has developed various architectural innovations to enhance structural understanding capabilities. Subgraph-based methods [81, 20, 80, 32] extract features from local structural patterns around nodes. Graph transformers [16, 39, 74, 50] enable broader context aggregation through global attention mechanisms. Particularly relevant to our work are positional encodings (PE) that augment graph models with pre-computed global information. Spectral approaches [17, 39] leverage Laplacian eigenvectors to encode global connectivity patterns. SPE [31] learns continuous soft-partition mappings that weight each eigenvector by its eigenvalue, ensuring both provable Lipschitz stability under graph perturbations and universal expressivity for basis-invariant functions. SignNet [41] addresses the ambiguity of eigenvectors by computing node-wise features through a learned function, followed by MLPs to produce sign-invariant representations. These methods demonstrate that injecting pre-computed global structural information can substantially improve performance on tasks requiring holistic graph understanding.

# C    Benchmark Statistics

Table 4 provides a comprehensive overview of our benchmark statistics across all four tasks. Each task is carefully designed with appropriate sample sizes and node ranges to enable robust evaluation. To gain deeper insights into the distribution characteristics of our regression tasks, we visualized the distributions of bridge counts and spectral gaps across different graph types and dataset splits in Figures 4 and 5.

Table 4: Dataset statistics across our four benchmark tasks. Each cell shows the number of graphs followed by the node count range in parentheses. Topology Classification is a 6-way classification task, Symmetry Classification is a 2-way classification task, while Spectral Gap and Bridge Count are regression tasks.

| Split | Topology | Symmetry | Spectral Gap | Bridge Count |
|---|---|---|---|---|
| **Train** | 3000 (20-50) | 2000 (30-60) | 3000 (20-50) | 2500 (20-50) |
| **Val** | 300 (20-50) | 200 (30-60) | 300 (20-50) | 250 (20-50) |
| **Test (ID)** | 300 (20-50) | 600 (30-60) | 300 (20-50) | 250 (20-50) |
| **Test (Near-OOD)** | 300 (40-100) | 600 (50-100) | 300 (40-100) | 250 (40-100) |
| **Test (Far-OOD)** | 300 (60-150) | 600 (70-150) | 300 (60-150) | 250 (60-150) |

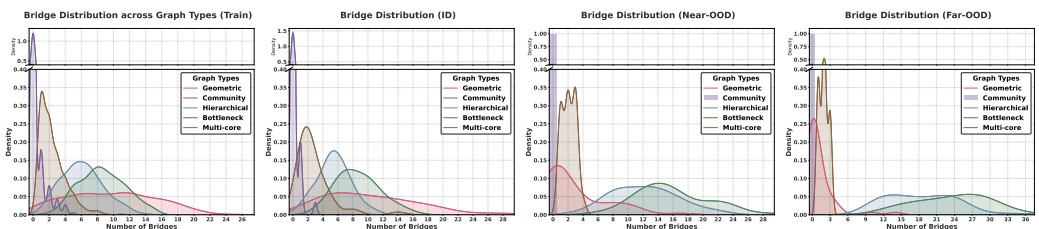

Figure 4: Distribution of bridge counts across different graph types under various settings (Train, ID, Near-OOD, and Far-OOD). The plots reveal distinct bridge count patterns for each graph structure (Geometric, Community, Hierarchical, Bottleneck, and Multicore). Notably, the distributions exhibit shifts as graph sizes increase, particularly visible in the OOD scenarios.

# D Implementation Details

This appendix provides comprehensive implementation details for our benchmark tasks, including dataset generation methodology, model architectures, and training protocols.

## D.1 Dataset Generation

Our benchmark consists of four distinct graph understanding tasks. For each task, we generate three test sets with varying difficulty: **ID** (same distribution as training), **Near-OOD** (moderate distribution shift), and **Far-OOD** (significant distribution shift). All datasets are implemented using PyTorch Geometric.

### D.1.1 Topology Classification Dataset Generation

For the topology classification task, we implemented six distinct topology generators, each producing graphs with visually and structurally distinct patterns.

**Cyclic Structure.** We generate annular random geometric graphs (ARGG) with inner radius randomly sampled from $[0.7, 1.2]$, outer radius set as $r_{inner} + \text{random}(0.05, 0.3)$, and connection radius randomly sampled from $[0.5, 0.8]$.

**Random Geometric Graph.** Nodes are distributed uniformly in a unit square with connection radius randomly sampled from $[0.15, 0.25]$. Connections are established between nodes within this radius of each other. We ensure connectedness by adding edges between disconnected components if necessary.

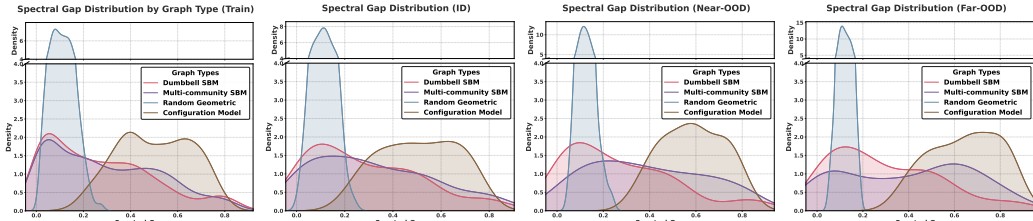

Figure 5: Distribution of spectral gaps across different graph types under various settings (Train, ID, Near-OOD, and Far-OOD). The plots reveal distinct spectral gap patterns for each graph structure. Notably, the distributions exhibit shifts as graph sizes increase.

**Community Structure.** We generate communities with 3 to 5 groups (based on graph size), intra-community edge probability from $[0.6, 0.8]$, and inter-community edge probability from $[0.01, 0.05]$. This creates densely connected communities with sparse connections between them, reflecting real-world community structures.

**Hierarchical Hub Structure.** We implement a multi-level hierarchical organization with 2 to 4 levels. Level size ratio decreases by a factor of approximately $0.4$ with higher levels. Intra-level connectivity decreases for lower levels, calculated as $0.7 \times \frac{\text{num\_levels}-\text{level}}{\text{num\_levels}}$. Each node connects to 1-3 nodes in the level above it, creating graphs with clear hierarchical organization from top to bottom.

**Bottleneck Structure.** We generate bottleneck topologies with 2 to 4 communities, intra-community edge probability from $[0.4, 0.6]$, and bottleneck width of 1 to 3nodes connecting adjacent communities. This creates graphs with distinct modules connected by narrow pathways.

**Multi-core Periphery Structure.** We generate multi-core networks with 2 to 3 cores occupying 50-60% of total nodes (minimum 4 nodes per core). Core internal probability ranges from $[0.6, 0.8]$ with 1-2 bridging nodes between each core pair. Periphery nodes are randomly connected to 1-2 nodes in a randomly selected core, creating structures with multiple highly connected centers surrounded by sparsely connected peripheral nodes.

### D.1.2 Symmetry Classification Dataset Generation

For symmetry classification, we employ several theoretically grounded approaches to generate both symmetric and asymmetric graphs.

**Real-world Base Graph Extraction.** To enhance diversity and realism in our generated graphs, we extracted a collection of base graphs from real-world datasets. For MUTAG, we directly utilized the molecular graphs. For Cora, which is a large citation network, we employed three different sampling techniques to obtain a diverse set of subgraphs: **(1)** BFS sampling from random starting nodes. **(2)** Random walks with restart probability $\alpha = 0.2$, which were explored by previous work [55]. **(3)** GraphSAGE-style neighborhood sampling with controlled layer expansion. For each target node size range (5-50 nodes), we ensured a sufficient number of base graphs (approximately 30 graphs per size) to support subsequent operations like Cartesian products and graph covers. These real-world base graphs were cached and reused across different generation methods, providing consistent structural patterns while maintaining diversity. All base graphs were processed to ensure connectedness, and their node indices were normalized to ensure compatibility with our generation pipelines.

We implemented five distinct methods to generate symmetric graphs:

**Cayley Cyclic Graphs.** We construct Cayley graphs based on cyclic groups $\mathbb{Z}_n$. For a given node count $n$, we identify valid generators (elements coprime to $n$), randomly select 1-3 generators, and create connections according to the Cayley graph definition. Each node $i \in \mathbb{Z}_n$ is connected to nodes $(i + g) \bmod n$ and $(i - g) \bmod n$ for each generator $g$, ensuring that the graph exhibits the algebraic symmetry of the cyclic group.

**Bipartite Double Cover.** We create bipartite double covers from various base graphs (random graphs, community structures, bottleneck structures, and real-world data). For a base graph $\mathcal{G} = (V, E)$, the bipartite double cover $\tilde{\mathcal{G}} = (\tilde{V}, \tilde{E})$ is constructed with $\tilde{V} = V \times {0, 1}$ and edges $\tilde{E} = {((u, 0), (v, 1)), ((u, 1), (v, 0)) \mid (u, v) \in E}$. Base graphs have approximately half the target node

count. For random base graphs, edge probability is from $[0.15, 0.3]$. For community-based graphs, intra-community probability is from $[0.3, 0.7]$ and inter-community probability is from $[0.05, 0.15]$. For bottleneck base graphs, internal connectivity probability is from $[0.3, 0.6]$ with bottleneck width of 1-3 nodes.

**Cartesian Product.** We generate cartesian products of known symmetric components with combinations including cycle graphs $C_n \square P_m$ (cycle $\square$ path), $C_n \square C_m$ (cycle $\square$ cycle), and $P_n \square S_m$ (path $\square$ star). The cartesian product $G_1 \square G_2$ contains vertices $V(\mathcal{G}_1) \times V(\mathcal{G}_2)$ with edges between $(u_1, v_1)$ and $(u_2, v_2)$ if either $u_1 = u_2$ and $(v_1, v_2) \in E(G_2)$, or $v_1 = v_2$ and $(u_1, u_2) \in E(G_1)$. Factors are chosen to optimize the product size to be close to the target node count.

**Cartesian Product with Real Data.** We compute Cartesian products using real-world graph data, selected from the MUTAG and Cora datasets. It is important to note that the Cartesian product preserves symmetry only when both base graphs are symmetric. Since real-world graphs typically lack perfect symmetry, we use the `pynauty`[1] library to verify the symmetry of each generated graph and filter accordingly. The introduction of real-world graphs enables us to generate more diverse and structurally realistic graphs, including both symmetric and asymmetric variants with complex topological features that purely synthetic generators cannot easily produce.

**Real Data Cyclic Cover.** We construct $k$-fold cyclic covers using real data as base graphs, with the number of layers ($k$) from 2 to 5 based on target size. For a base graph $G = (V, E)$, the $k$-fold cyclic cover creates $k$ copies of $V$ with edges connecting corresponding vertices across consecutive layers in a cyclic pattern. Formally, vertices in the cover are $V \times \mathbb{Z}_k$, and for each edge $(u, v) \in E$, we add edges $((u, i), (v, (i + 1) \bmod k))$ and $((v, i), (u, (i + 1) \bmod k))$ for all $i \in \mathbb{Z}_k$.

We implemented two primary approaches to generate asymmetric graphs:

**Perturbed Asymmetric Graphs.** We start with symmetric graphs and apply targeted edge perturbations via double-edge swap operations. We select two edges $(a, b)$ and $(c, d)$ and replace them with edges $(a, d)$ and $(c, b)$ if they don't already exist. This operation maintains the degree distribution while potentially breaking symmetry. We verify that connectivity is preserved and symmetry is broken, applying up to 20 swap attempts. Each generated graph is verified to ensure $|\text{Aut}(G)| = 1$, meaning the only automorphism is the identity mapping.

**Cartesian Products with Real Graphs.** We leverage the inherent asymmetry of real-world data through Cartesian products involving the previously extracted real-world networks. It's important to note that while the Cartesian product operation naturally creates repeating structural patterns (as each vertex from one graph is combined with every vertex from the other graph), these repeating patterns do not necessarily translate into mathematical symmetry (automorphisms). The resulting structure may contain similar local neighborhoods but still lack the global permutation invariance required for true symmetry. This distinction is crucial - Cartesian products create structural regularity that can be challenging for models to analyze, but often without introducing the simplifying symmetry properties that might make the task easier. All generated graphs are rigorously verified using `pynauty` to confirm their asymmetric nature ($|\text{Aut}(G)| = 1$) before inclusion in the dataset.

### D.1.3 Spectral Gap Regression

For spectral gap regression, we employed a mixture of techniques to generate graphs with diverse spectral properties:

**SBM Evolution.** Using Stochastic Block Models(SBM) with controlled mixing, including SBM-dumbbell structure (two equal-sized communities) and SBM-multi communities structure (3-5 communities). Within-block probability ranges from $[0.6, 0.8]$. Between-block probability varies with mixing parameter $\mu \in [0, 1]$: $[0.001, 0.02]$ when $\mu < 0.1$, $[0.02, 0.1]$ when $\mu < 0.3$, $[0.1, 0.3]$ when $\mu < 0.7$, and $[0.3, \mu]$ when $\mu \geq 0.7$. The spectral gap $\lambda_2$ of these graphs is strongly influenced by the between-block connectivity.

**Geometric Evolution.** Modified random geometric graphs with base radius calculated to ensure basic connectivity. Additional connections are added based on mixing parameter $\mu$, with the number of extra connections approximately equal to $\mu \times 0.1 \times n \times \ln(n)$. Higher $\mu$ values create more small-world-like properties, affecting the spectral gap.

---

[1] https://github.com/pdobsan/pynauty

**Configuration Model.** Starting with an SBM or geometric base graph, we rewire edges using the configuration model while preserving the degree distribution. The randomization level controls the fraction of edges rewired, ranging from $[0.3, 0.8]$. This process disrupts the original structure while maintaining the degree sequence, often resulting in graphs with different spectral properties.

To ensure comprehensive coverage of the spectral gap value range, we strategically sample the mixing parameter $\mu$, with $40\%$ of samples using low connectivity ($\mu < 0.2$), $30\%$ using medium connectivity ($\mu \in [0.2, 0.5]$), and $30\%$ using higher connectivity ($\mu \in [0.5, 0.8]$). The spectral gap $\lambda_2$ is computed as the second-smallest eigenvalue of the normalized Laplacian matrix $\mathcal{L} = I - D^{-1/2}AD^{-1/2}$, where $D$ is the degree matrix and $A$ is the adjacency matrix.

### D.1.4 Bridge Counting Dataset Generation

For the bridge counting task, we generate diverse graph structures using five topology generators similar to those used in the topology classification task. These include Random Geometric Graphs with connection radius $r \in [0.15, 0.25]$, Community Structures with 3-5 communities and controlled density parameters, Hierarchical Structures with 2-4 levels, Bottleneck Configurations with single-connection bottlenecks between adjacent communities, and Multi-core Structures with 2-3 cores and peripheral attachments. For each graph, we compute the exact number of bridges using `NetworkX`'s bridge detection algorithm[2], which identifies edges whose removal would increase the number of connected components in the graph.

## D.2 Node Features and Positional Encodings

All positional encodings are incorporated into our GNN models with 16 dimensions. For node features, we use one-hot degree encoding with a maximum degree of 100 across all graph datasets.

## D.3 Model Architecture and Hyperparameters

For our graph neural network models, we experiment with varying numbers of layers ranging from 2 to 4, with a consistent hidden dimension size of 128 across all architectures. Dropout with a rate of 0.5 is applied throughout the networks to prevent overfitting. For vision-based models, we use standard architectures: ResNet-50, ViT-B/16, Swin Transformer-Tiny, and ConvNeXtV2-Tiny. All models resize graph images to 224×224 resolution as input.

## D.4 Training Protocol

All models are trained with a batch size of 128 for a maximum of 200 epochs, employing early stopping with a patience of 30 epochs to prevent overfitting. We use the Adam optimizer with different learning rates: $1e-5$ for vision backbone parameters, $1e-3$ for GNN models and classifier heads. Weight decay is set to $1e-4$ for vision models. For classification tasks (Topology, Symmetry), we use cross-entropy loss, while for regression tasks (Spectral Gap, Bridge Counting), we employ mean squared error loss. All experiments are conducted on 4 NVIDIA A800 GPUs. For consistent evaluation, we measure accuracy for classification tasks, while regression tasks use Mean Absolute Error (MAE). To ensure reproducibility, we set fixed random seeds $\in [0, 1, 2, 3, 4]$ for all experiments, controlling the initialization of model parameters, data splitting.

## D.5 Graph Image Generation for Vision Models

For vision-based models, we render graph visualizations with specific parameters to ensure visual consistency. Nodes are rendered as skyblue circles with white borders and size 50, while edges are rendered as white lines with width 1.5 and alpha 0.8. Each graph is rendered once for each dataset split, ensuring consistent visual representation across training and evaluation.

---

[2]https://networkx.org

# E   Proof of Symmetry in Graph Coverings

In this section, we provide formal proofs of the symmetry properties stated in the main text. For notational simplicity, we use $H$ to denote a cover graph throughout these proofs. This corresponds to $\tilde{G}$ (bipartite double cover) and $G_k$ ($k$-fold cyclic cover) in the main text.

## E.1   Definitions

**Definition 1** (Automorphism). *An automorphism of a graph $\mathcal{G}$ is a bijection $\phi : V(\mathcal{G}) \to V(\mathcal{G})$ such that for any $u, v \in V(\mathcal{G})$:*

$$(u, v) \in E(\mathcal{G}) \iff (\phi(u), \phi(v)) \in E(\mathcal{G}) \tag{7}$$

**Definition 2** (Automorphism Group). *The set of all automorphisms of $\mathcal{G}$, denoted $\mathrm{Aut}(\mathcal{G})$, forms a group under function composition.*

**Definition 3** (Symmetry). *A graph $\mathcal{G}$ is symmetric if $|\mathrm{Aut}(\mathcal{G})| > 1$, i.e., it admits at least one non-identity automorphism.*

**Definition 4** (Bipartite Double Cover). *For a graph $\mathcal{G}$, its bipartite double cover $H$ is constructed by:*

- *Creating two vertices $(v, 0)$ and $(v, 1)$ in $H$ for each vertex $v$ in $\mathcal{G}$*

- *Creating edges $((u, 0), (v, 1))$ and $((u, 1), (v, 0))$ in $H$ for each edge $(u, v)$ in $\mathcal{G}$*

**Definition 5** ($k$-fold Cyclic Cover). *For a graph $\mathcal{G}$ (without self-loops), its $k$-fold cyclic cover $H$ is constructed by:*

- *Creating $k$ vertices $(v, 0), (v, 1), \ldots, (v, k-1)$ in $H$ for each vertex $v$ in $G$*

- *Creating edges $((u, i), (v, (i+1) \bmod k))$ and $((v, i), (u, (i+1) \bmod k))$ in $H$ for each edge $(u, v)$ in $G$ and each $i \in \{0, 1, \ldots, k-1\}$*

## E.2   Symmetry of Bipartite Double Cover

**Theorem 1.** *For any graph $\mathcal{G}$, its bipartite double cover $H$ satisfies $|\mathrm{Aut}(H)| > 1$.*

*Proof.* We define a mapping $\sigma : V(H) \to V(H)$ as follows:

$$\sigma((v, i)) = (v, 1 - i) \quad \forall v \in V(\mathcal{G}), i \in \{0, 1\} \tag{8}$$

**Step 1:** We prove $\sigma$ is a bijection.
Since for each $(v, i) \in V(H)$, $\sigma$ maps to exactly one element $(v, 1 - i) \in V(H)$, and since $\sigma(\sigma((v, i))) = \sigma((v, 1 - i)) = (v, i)$, $\sigma$ is its own inverse. Therefore, $\sigma$ is a bijection.

**Step 2:** We verify $\sigma$ preserves edges.
For any edge $e = ((u, i), (v, j)) \in E(H)$, by the definition of double cover:

$$(u, v) \in E(\mathcal{G}) \text{ and } j = 1 - i \tag{9}$$

Applying $\sigma$ to both endpoints:

$$\sigma((u, i)) = (u, 1 - i) \text{ and } \sigma((v, j)) = \sigma((v, 1 - i)) = (v, i) \tag{10}$$

Since $(u, v) \in E(\mathcal{G})$, by the definition of double cover:

$$((u, 1 - i), (v, i)) \in E(H) \tag{11}$$

Thus, $(\sigma(u, i), \sigma(v, j)) \in E(H)$.

**Step 3:** We verify $\sigma$ preserves non-edges.
For any non-edge $((u, i), (v, j)) \notin E(H)$, there are two cases:

1. $(u, v) \notin E(\mathcal{G})$: Then $((u, 1 - i), (v, 1 - j)) \notin E(H)$ by definition of double cover.

2. $(u, v) \in E(\mathcal{G})$ but $j \neq 1 - i$: Then either $i = j = 0$ or $i = j = 1$. After applying $\sigma$, we have $\sigma((u, i)) = (u, 1 - i)$ and $\sigma((v, j)) = (v, 1 - j)$. Since $1 - i = 1 - j$, we have $(\sigma(u, i), \sigma(v, j)) \notin E(H)$.

**Step 4:** We show $\sigma$ is not the identity mapping.
For any vertex $(v, 0) \in V(H)$:

$$\sigma((v, 0)) = (v, 1) \neq (v, 0) \tag{12}$$

Therefore, $\sigma$ is a non-identity automorphism of $H$, which proves $|\mathrm{Aut}(H)| > 1$. □

**Corollary 1.** *If* $|\mathrm{Aut}(\mathcal{G})| = m$, *then* $|\mathrm{Aut}(H)| \geq 2m$.

*Proof.* Every automorphism $\phi \in \mathrm{Aut}(\mathcal{G})$ induces two automorphisms on $H$:

$$\phi_1((v, i)) = (\phi(v), i) \quad \text{and} \quad \phi_2((v, i)) = (\phi(v), 1 - i) \tag{13}$$

These $2m$ automorphisms are all distinct, hence $|\mathrm{Aut}(H)| \geq 2m$. □

## E.3 Symmetry of $k$-fold Cyclic Cover

**Lemma 1.** *For* $k \geq 2$, *the map* $\tau : V(H) \rightarrow V(H)$ *defined by* $\tau((v, i)) = (v, (i + 1) \bmod k)$ *is an automorphism of the $k$-fold cyclic cover $H$ of any graph $\mathcal{G}$.*

*Proof.* **Step 1:** We prove $\tau$ is a bijection.
For each $(v, i) \in V(H)$, $\tau$ maps to exactly one element $(v, (i + 1) \bmod k)$. The inverse is defined by $\tau^{-1}((v, i)) = (v, (i - 1) \bmod k)$. Hence, $\tau$ is a bijection.

**Step 2:** We verify $\tau$ preserves edges.
Let $e = ((u, i), (v, (i + 1) \bmod k)) \in E(H)$. By definition of $k$-fold cover, $(u, v) \in E(\mathcal{G})$.

Applying $\tau$ to both endpoints:

$$\begin{aligned}
\tau((u, i)) &= (u, (i + 1) \bmod k) \\
\tau((v, (i + 1) \bmod k)) &= (v, (i + 2) \bmod k)
\end{aligned} \tag{14}$$

Since $(u, v) \in E(\mathcal{G})$, by definition of $k$-fold cover:

$$((u, (i + 1) \bmod k), (v, (i + 2) \bmod k)) \in E(H) \tag{15}$$

**Step 3:** We verify $\tau$ preserves non-edges.
For any non-edge $((u, i), (v, j)) \notin E(H)$, there are two cases:

1. $(u, v) \notin E(\mathcal{G})$: Then $((u, (i + 1) \bmod k), (v, (j + 1) \bmod k)) \notin E(H)$ by definition.

2. $(u, v) \in E(\mathcal{G})$ but $j \neq (i + 1) \bmod k$: After applying $\tau$, we have

$$\begin{aligned}
\tau((u, i)) &= (u, (i + 1) \bmod k) \\
\tau((v, j)) &= (v, (j + 1) \bmod k)
\end{aligned} \tag{16}$$

Since $(j + 1) \bmod k \neq (i + 2) \bmod k$ when $j \neq (i + 1) \bmod k$, we have $(\tau(u, i), \tau(v, j)) \notin E(H)$.

**Step 4:** For $k \geq 2$, $\tau$ is not the identity mapping.
For any vertex $(v, 0) \in V(H)$:

$$\tau((v, 0)) = (v, 1) \neq (v, 0) \tag{17}$$

Therefore, $\tau$ is a non-identity automorphism of $H$. □

**Theorem 2.** *For any graph $\mathcal{G}$ and $k \geq 2$, the $k$-fold cyclic cover $H$ admits an automorphism group containing a subgroup isomorphic to $\mathbb{Z}_k$, thus satisfying* $|\mathrm{Aut}(H)| \geq k$.

*Proof.* Consider the mappings $\tau, \tau^2, \ldots, \tau^{k-1}, \tau^k$, where $\tau$ is the automorphism defined in the lemma above. We have:

$$\tau^j((v,i)) = (v, (i+j) \bmod k) \tag{18}$$

**Step 1:** The mappings $\tau^0 = \mathrm{id}, \tau^1, \tau^2, \ldots, \tau^{k-1}$ are all distinct.
For $0 \le j_1 < j_2 < k$, there exists $(v,0) \in V(H)$ such that:

$$\tau^{j_1}((v,0)) = (v, j_1) \ne (v, j_2) = \tau^{j_2}((v,0)) \tag{19}$$

**Step 2:** These mappings form a cyclic subgroup of order $k$.
Since $\tau^k((v,i)) = (v, (i+k) \bmod k) = (v,i)$, we have $\tau^k = \mathrm{id}$. This means $\tau$ generates a cyclic group of order $k$ isomorphic to $\mathbb{Z}_k$.

Therefore, $\mathrm{Aut}(H)$ contains a subgroup isomorphic to $\mathbb{Z}_k$, implying $|\mathrm{Aut}(H)| \ge k$. □

### E.4 Algorithmic Implementation

The implementation of graph coverings in our code precisely follows the mathematical constructions in the above definitions:

---

**Algorithm 1** Generate Bipartite Double Cover

---

**Require:** Base graph $\mathcal{G}(V,E)$
**Ensure:** Double cover graph $H$
 1: Initialize $H$ as empty graph
 2: **for** each $v \in V$ **do**
 3:     Add vertices $(v,0)$ and $(v,1)$ to $H$
 4: **end for**
 5: **for** each $(u,v) \in E$ **do**
 6:     Add edges $((u,0),(v,1))$ and $((u,1),(v,0))$ to $H$
 7: **end for**
 8: **return** $H$

---

**Algorithm 2** Generate $k$-fold Cyclic Cover from Real-world Network

---

**Require:** Real-world base graph $\mathcal{G}(V,E)$, integer $k \ge 2$
**Ensure:** $k$-fold cyclic cover graph $H$
 1: Initialize $H$ as empty graph
 2: **for** each $v \in V$ **do**
 3:     **for** $i = 0$ to $k - 1$ **do**
 4:         Add vertex $(v,i)$ to $H$
 5:     **end for**
 6: **end for**
 7: **for** each $(u,v) \in E$ **do**
 8:     **for** $i = 0$ to $k - 1$ **do**
 9:         Add edges $((u,i),(v,(i+1) \bmod k))$ and $((v,i),(u,(i+1) \bmod k))$ to $H$
10:     **end for**
11: **end for**
12: **return** $H$

---

# F    Extended Experimental Results

This section provides detailed experimental results that support our main findings, including comprehensive computational cost analysis, model scaling experiments, advanced GNN comparisons, and resolution robustness studies.

**Computational Cost Analysis.** Table 5 presents a detailed breakdown of computational requirements across all four benchmark tasks. We report both time per epoch and total time to reach best validation accuracy. Vision models require approximately 10× more time per epoch than GNN+SPE models across all tasks, with the ratio ranging from 8.2× to 12.2× depending on the specific task. This computational overhead primarily stems from the larger model capacity and more complex feature extraction in vision backbones compared to compact GNN architectures.

**Model Scaling Analysis.** To investigate whether performance differences stem from model capacity, we conducted systematic scaling experiments with GPS+SPE. Table 6 shows results for models with 53.2M and 212.2M parameters, substantially exceeding vision model sizes (ResNet-50: 25.6M). Consistent with known limitations of message-passing architectures, performance degraded rather than improved with increased capacity across all tasks. For topology classification, accuracy dropped from 84.80% (baseline) to 82.53% (212.2M model) on Near-OOD tasks. This confirms that architectural differences, rather than parameter count, drive the observed advantages of vision models.

**Comparison with Advanced GNNs.** Table 7 compares our results with I²-GNN [32], an advanced structure-aware GNN. While I²-GNN achieves competitive in-distribution performance, it shows significantly worse generalization on out-of-distribution tasks, particularly on Far-OOD settings. This demonstrates that even advanced GNN architectures with enhanced expressiveness face challenges in scale generalization compared to vision-based approaches.

**Resolution Robustness.** Table 8 examines how image resolution affects vision model performance. Performance remains relatively stable across different resolutions (64×64 to 448×448), with some tasks even performing better at lower resolutions. This suggests that the structural patterns captured by vision models are robust to resolution changes, and extremely high resolution may not be necessary for graph structural understanding tasks.

Table 5: Computational cost analysis across all benchmark tasks, showing time per epoch and time to best validation accuracy.

| Model | Topology | | Symmetry | | Bridge | | Spectral Gap | |
|---|---|---|---|---|---|---|---|---|
| | Time/Epoch (s) | Time to Best (s) | Time/Epoch (s) | Time to Best (s) | Time/Epoch (s) | Time to Best (s) | Time/Epoch (s) | Time to Best (s) |
| **ConvNeXt** | 13.1 | 329.5 | 9.4 | 245.2 | 11.2 | 573.6 | 13.2 | 717.7 |
| **ResNet** | 5.5 | 120.7 | 4.2 | 76.5 | 4.8 | 127.5 | 5.6 | 312.0 |
| **Swin** | 11.5 | 368.9 | 8.2 | 123.1 | 9.8 | 227.1 | 11.6 | 348.2 |
| **ViT** | 23.1 | 509.9 | 15.8 | 127.0 | 19.5 | 858.6 | 23.2 | 1139.7 |
| **GAT+SPE** | 1.2 | 43.2 | 0.9 | 16.3 | 1.1 | 2.5 | 1.2 | 107.7 |
| **GCN+SPE** | 1.2 | 45.1 | 0.8 | 5.3 | 1.0 | 93.3 | 1.2 | 35.1 |
| **GIN+SPE** | 1.1 | 47.0 | 0.8 | 6.1 | 1.1 | 56.0 | 1.2 | 30.6 |
| **GPS+SPE** | 1.4 | 26.9 | 1.0 | 22.0 | 1.2 | 34.9 | 1.5 | 32.7 |
| Avg. Vision | 13.3 | 332.2 | 9.4 | 142.9 | 11.3 | 446.7 | 13.4 | 629.4 |
| Avg. GNN+SPE | 1.2 | 40.6 | 0.9 | 12.4 | 1.1 | 46.7 | 1.3 | 51.5 |
| **Ratio (V/G)** | **10.9×** | **8.2×** | **10.7×** | **11.5×** | **10.3×** | **9.6×** | **10.5×** | **12.2×** |

Table 6: Model scaling analysis across all benchmark tasks. Scaled GPS+SPE models show degraded performance compared to baseline, confirming that architectural constraints rather than parameter count limit GNN performance.

| Task | Model | ID | Near-OOD | Far-OOD |
|---|---|---|---|---|
| **Topology (%)** | Baseline GPS+SPE | 84.80 ± 13.75 | 84.07 ± 16.04 | 72.20 ± 14.51 |
| | GPS+SPE (53.2M) | 87.73 ± 6.87 | 70.13 ± 13.03 | 40.53 ± 11.06 |
| | GPS+SPE (212.2M) | 82.53 ± 5.58 | 66.20 ± 11.42 | 34.80 ± 8.16 |
| | ResNet (25.6M) | **95.87 ± 0.62** | **96.27 ± 1.02** | **87.40 ± 3.33** |
| **Symmetry (%)** | Baseline GPS+SPE | 71.97 ± 1.65 | 70.67 ± 1.23 | 67.70 ± 1.37 |
| | GPS+SPE (53.2M) | 65.83 ± 2.21 | 65.83 ± 2.83 | 67.63 ± 3.94 |
| | GPS+SPE (212.2M) | 56.10 ± 4.47 | 55.67 ± 5.75 | 54.97 ± 4.53 |
| | ResNet (25.6M) | **93.47 ± 0.66** | **88.83 ± 0.64** | **84.20 ± 0.39** |
| **Spectral Gap (MAE)** | Baseline GPS+SPE | 0.0681 ± 0.0298 | 0.1537 ± 0.0839 | 0.6716 ± 0.2709 |
| | GPS+SPE (53.2M) | 0.1483 ± 0.0210 | 0.1901 ± 0.0167 | 0.7497 ± 0.4243 |
| | GPS+SPE (212.2M) | 0.1214 ± 0.0365 | 0.2125 ± 0.0373 | 0.9101 ± 0.5625 |
| | ResNet (25.6M) | **0.0335 ± 0.0021** | **0.0600 ± 0.0063** | **0.1102 ± 0.0100** |
| **Bridge Count (MAE)** | Baseline GPS+SPE | 0.6402 ± 0.1753 | 1.4666 ± 0.0713 | 3.8021 ± 1.0492 |
| | GPS+SPE (53.2M) | 1.4502 ± 0.2315 | 3.0053 ± 0.5616 | 5.6101 ± 0.8129 |
| | GPS+SPE (212.2M) | 2.1581 ± 0.8106 | 3.3334 ± 0.7540 | 5.7994 ± 1.1003 |
| | ResNet (25.6M) | **0.7771 ± 0.1095** | **1.6356 ± 0.1643** | **3.6814 ± 0.1217** |

Table 7: Comparison with I²-GNN across all benchmark tasks. I²-GNN shows strong symmetry detection but poor scale generalization.

| Task | Model | ID | Near-OOD | Far-OOD |
|---|---|---|---|---|
| **Topology** (%) | I²-GNN | 94.67 ± 1.49 | 87.87 ± 4.90 | 63.33 ± 5.83 |
| | Best Vision | **95.87 ± 0.62** | **97.73 ± 0.57** | **90.33 ± 4.60** |
| **Symmetry** (%) | I²-GNN | 90.80 ± 3.49 | 84.00 ± 2.53 | 83.40 ± 3.56 |
| | Best Vision | **94.03 ± 1.04** | **91.03 ± 0.56** | **85.67 ± 1.06** |
| **Spectral** (MAE) | I²-GNN | 0.1044 ± 0.0091 | 0.3315 ± 0.0941 | 0.7893 ± 0.2709 |
| | Best Vision | **0.0279 ± 0.0047** | **0.0578 ± 0.0056** | **0.0946 ± 0.0094** |
| **Bridge** (MAE) | I²-GNN | **0.4580 ± 0.1131** | **1.0459 ± 0.1523** | 3.7353 ± 1.0961 |
| | Best Vision | 0.6261 ± 0.0702 | 1.6338 ± 0.1675 | **3.6814 ± 0.1217** |

Table 8: Impact of image resolution on vision model performance (ResNet on Symmetry classification).

| Resolution | ID | Near-OOD | Far-OOD |
|---|---|---|---|
| **Kamada-Kawai Layout** | | | |
| 448×448 | 80.87 ± 1.80 | 76.27 ± 2.99 | 72.60 ± 4.15 |
| 224×224 | 84.30 ± 0.36 | 81.07 ± 0.56 | **75.60 ± 2.19** |
| 128×128 | 84.50 ± 0.46 | **82.73 ± 0.90** | 74.77 ± 1.60 |
| 64×64 | **86.20 ± 0.69** | 81.03 ± 1.50 | 72.73 ± 2.22 |
| **Spectral Layout** | | | |
| 448×448 | **93.97 ± 0.96** | **91.53 ± 0.71** | **85.93 ± 0.23** |
| 224×224 | 93.17 ± 1.15 | 89.47 ± 0.46 | 83.20 ± 0.97 |
| 128×128 | 93.53 ± 0.69 | 88.30 ± 0.40 | 81.23 ± 0.88 |
| 64×64 | 91.97 ± 1.27 | 87.17 ± 1.48 | 80.93 ± 1.08 |

# G   Discussion: Visual vs. Message-Passing Paradigms

In this section, we provide an informal analysis of why vision models achieve strong performance on graph structural understanding despite lacking graph-specific inductive biases. While we lack formal theoretical proofs, several observations from our community's own research practices offer intuitions about the complementary computational mechanisms underlying these approaches.

**Different Computational Paradigms.** The two paradigms solve fundamentally different problems. GNNs take graph structure (adjacency matrices, edge lists) as input and build understanding through iterative local message passing. Layout algorithms, in contrast, perform global computations upfront: eigendecomposition for spectral layouts, energy minimization for force-directed methods. Once graphs are rendered as images, the task transforms from graph analysis to visual pattern recognition. Vision models then process geometric patterns where structural properties manifest as directly observable visual features: symmetric graphs produce symmetric layouts, clustered graphs show dense regional connections, bridges appear as narrow connectors between substructures.

**Evidence from Known GNN Limitations.** This difference becomes evident when examining tasks where GNNs face theoretical limitations. When researchers construct counterexamples for the Weisfeiler-Lehman test and its $k$-dimensional variants, they invariably use graph visualizations to illustrate why graphs are non-isomorphic despite fooling the WL algorithm [61]. Visual representations make structural distinctions immediately apparent that iterative refinement procedures miss. Horn et al. [29] explicitly present datasets containing graphs they describe as "*easily distinguished by humans*" visually. Their NECKLACES dataset shows graphs with identical cycle counts but different connectivity patterns: two individual cycles versus a merged one. While humans immediately see this difference, standard message-passing approaches fail to distinguish them, requiring sophisticated persistent homology calculations. Similarly, Zhang et al. [79] proved that standard GNNs cannot identify bridges, yet these structures manifest as obvious visual bottlenecks in graph layouts.

These observations suggest that layout algorithms and vision models provide a complementary pathway to graph understanding: layout algorithms convert abstract topological properties into spatial patterns through global computation, while vision models recognize these geometric features through hierarchical processing. This explains why vision models maintain strong performance despite lacking graph-specific inductive biases: they access structural information through a fundamentally different computational mechanism than message passing.

# H Visualization Examples from GraphAbstract

In this section, we provide visualizations of representative graphs in our benchmark. Figures 6–8 illustrate the diverse topological patterns across the four tasks: topology classification, symmetry classification, and spectral gap regression.

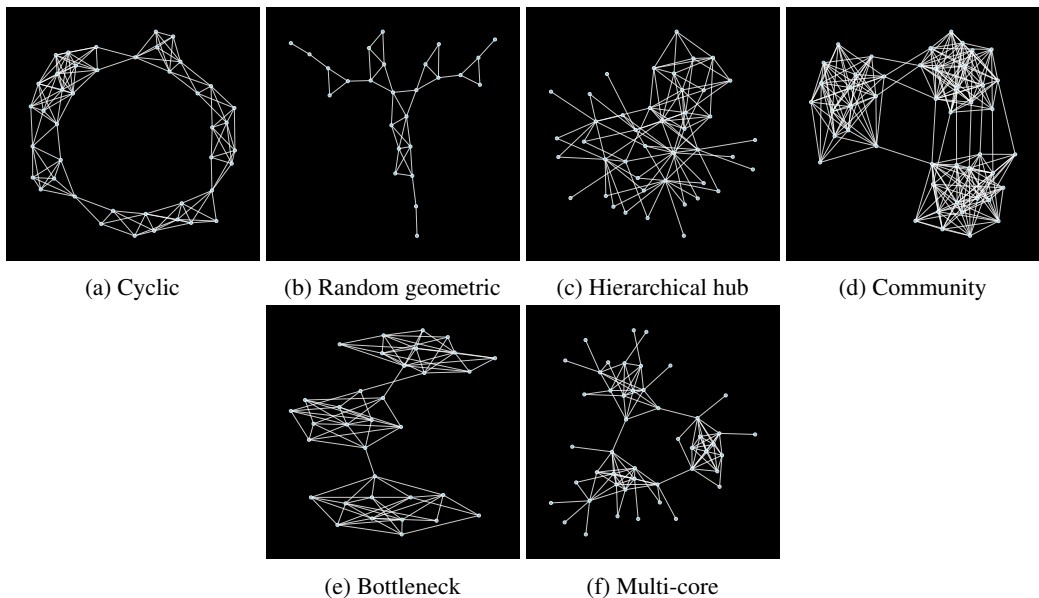

(a) Cyclic    (b) Random geometric    (c) Hierarchical hub    (d) Community

(e) Bottleneck    (f) Multi-core

Figure 6: Training examples of topology classification. For bridge counting task, similar graph structures are used except for the cyclic structure, as bridge counting focuses on identifying edges whose removal would disconnect the graph.

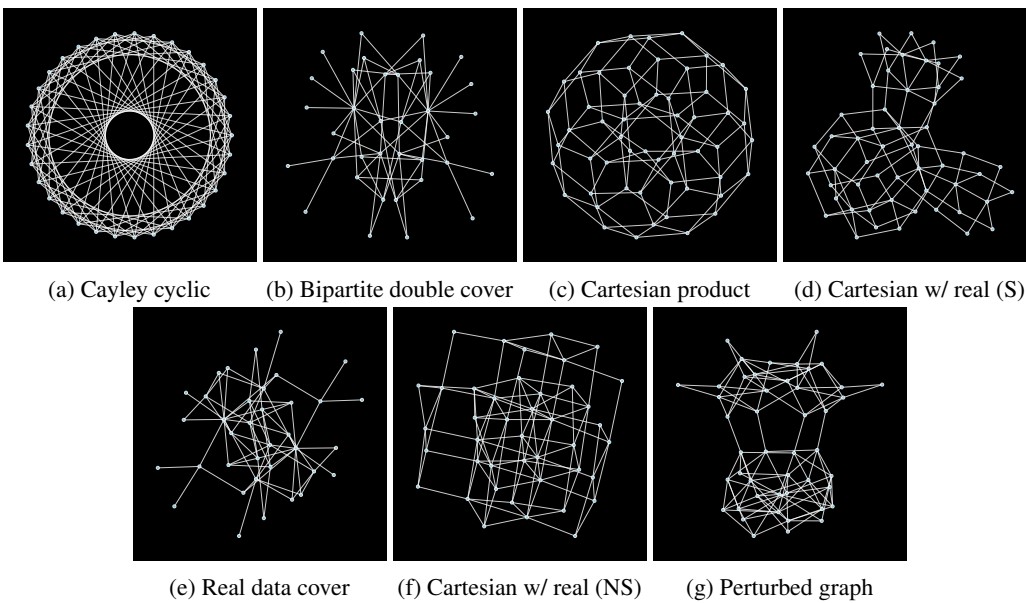

(a) Cayley cyclic    (b) Bipartite double cover    (c) Cartesian product    (d) Cartesian w/ real (S)

(e) Real data cover    (f) Cartesian w/ real (NS)    (g) Perturbed graph

Figure 7: Training examples of symmetric classification. (a)-(e) are symmetric graphs, while (f)-(g) are asymmetric graphs. S: symmetric, NS: non-symmetric.

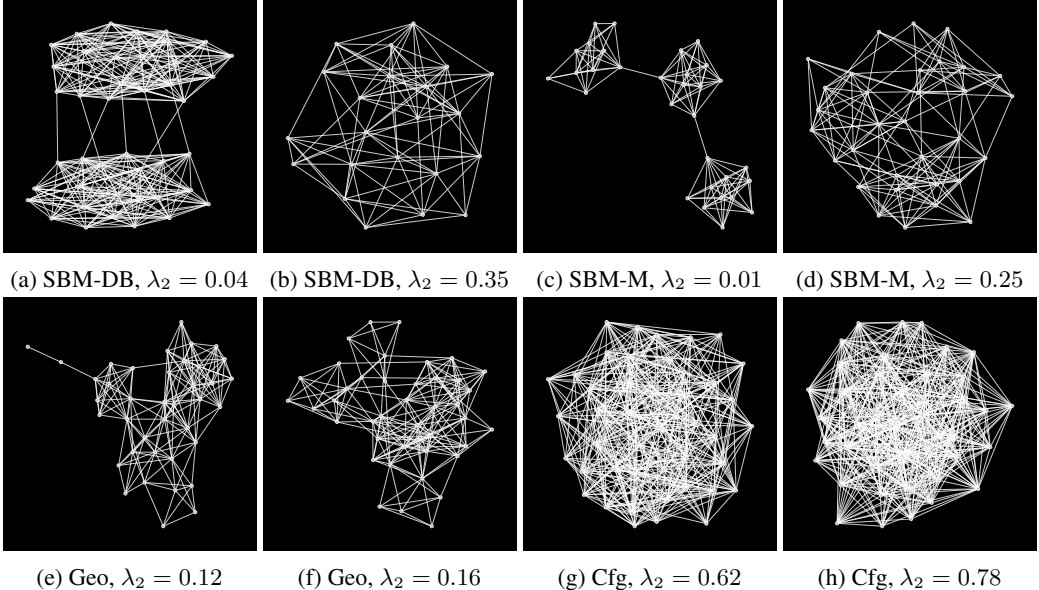

(a) SBM-DB, $\lambda_2 = 0.04$    (b) SBM-DB, $\lambda_2 = 0.35$    (c) SBM-M, $\lambda_2 = 0.01$    (d) SBM-M, $\lambda_2 = 0.25$

(e) Geo, $\lambda_2 = 0.12$    (f) Geo, $\lambda_2 = 0.16$    (g) Cfg, $\lambda_2 = 0.62$    (h) Cfg, $\lambda_2 = 0.78$

Figure 8: Training examples of spectral gap regression, where $\lambda_2$ is the second smallest eigenvalue of the normalized Laplacian. SBM-DB: SBM dumbbell, SBM-M: SBM multi-community, Geo: Geometric, Cfg: Configuration model.

# I  Analysis of Graph Layout Algorithms

## I.1  Layout Algorithm Details

Graph layout algorithms aim to produce 2D (or 3D) representations of graphs that are interpretable and reveal underlying structures. Different algorithms employ distinct heuristics and optimization criteria, leading to varied visual outputs. In this work, we primarily consider four common layout types:

**Kamada-Kawai.** This is a force-directed algorithm that models the graph as a system of springs. It aims to position nodes such that the geometric distance between them in the layout is proportional to their graph-theoretic distance (shortest path length). This often results in aesthetically pleasing layouts that emphasize the overall shape and connectivity.

**ForceAtlas2.** Another popular force-directed algorithm, particularly well-suited for larger graphs. It simulates attraction forces between connected nodes and repulsion forces between all nodes, often effectively revealing clusters and community structures within the network.

**Spectral Layout.** This method uses the eigenvectors of the graph Laplacian (or a related matrix) as coordinates for the nodes. Typically, the eigenvectors corresponding to the smallest non-zero eigenvalues are used. Spectral layouts are mathematically principled and often highlight global symmetries and Cheeger-type cuts. A characteristic feature is that structurally equivalent or highly similar nodes can overlap in the visualization.

**Circular Layout.** One of the simplest layout algorithms, it places all nodes equidistant on the circumference of a circle. The ordering of nodes around the circle can be arbitrary, based on node IDs, or determined by other properties like node degree.

## I.2  Visualizing Asymmetric Graphs

To illustrate how different layout algorithms can affect the visual perception of graph properties, particularly symmetry and structural regularity, we present visualizations of a non-symmetric graph in Figure 9. This graph is generated via the Cartesian product of two distinct real-world graphs, resulting in a structure with high local regularity but no global symmetry. As detailed in the caption, force-directed layouts (Kamada-Kawai and ForceAtlas2) tend to faithfully represent the resulting complex structure. Consequently, to determine its asymmetry, one might need to mentally deconstruct the layout to infer the properties of the underlying, distinct base graphs and recognize that their product would not yield simple visual symmetry. In contrast, the spectral and circular layouts render the graph such that its lack of global symmetry is more immediately visually evident.

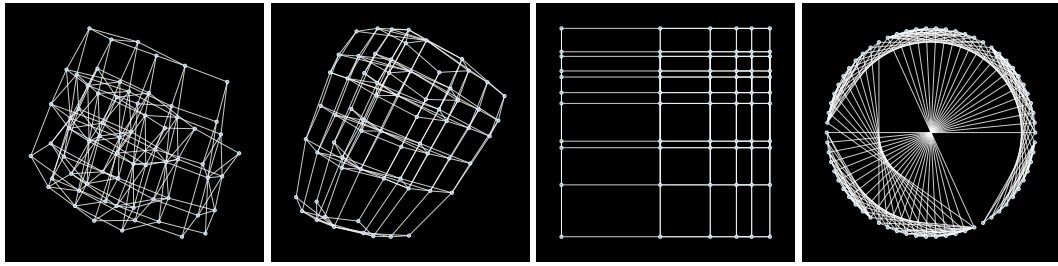

Figure 9: Visualizations of a non-symmetric graph generated by the Cartesian product of real-world base graphs. Despite its high structural regularity, this graph lacks symmetry. In Kamada-Kawai (**first**) and ForceAtlas2 (**second**) layouts, determining asymmetry requires mentally reconstructing the base graphs and analyzing their properties. In contrast, the Spectral layout (**third**) reveals the grid-like product structure. However, critical visual cues about its asymmetry are evident in the lack of perfect geometric regularity and the varying **thickness/brightness** of lines due to edge overlap. These imperfections prevent a global geometric symmetry axis (like a vertical line through the center) from mapping the graph onto itself, serving as clear visual signals of the underlying asymmetry. Circular layout (**fourth**) displays nodes on a circle and visually emphasizes **the lack of uniformity** in edge distribution, also clearly indicating the absence of symmetry. Both Spectral and Circular layouts facilitate a more direct visual assessment of asymmetry.

## I.3 Characteristics and Performance of Circular Layout

We present the performance of vision models using circular-layout generated graph visualization in Table 9. Despite circular layouts significantly disrupting many visual topological properties, evidenced by the sharp performance drop on Topology classification in Far-OOD settings, they prove remarkably effective for tasks like symmetry detection. This effectiveness stems from how symmetry (or its absence) becomes visually apparent: for symmetric graphs, edge densities are balanced across the circle, while for asymmetric graphs, humans can readily perceive uneven edge densities or specific edges that break the expected uniform pattern. This allows both humans and models to assess symmetry without needing to discern complex topological features that the circular layout inherently obscures.

Table 9: Performance comparison of circular layout across different tasks and models

| Task/Model | Swin | ConvNeXtV2 | ResNet | ViT |
|---|---|---|---|---|
| **Topology** | | | | |
| **ID** | **92.00 ± 0.56** | 90.89 ± 1.29 | **92.27 ± 1.00** | 91.47 ± 0.98 |
| **Near-OOD** | **81.13 ± 1.09** | 79.22 ± 2.01 | 75.40 ± 2.35 | **80.33 ± 2.80** |
| **Far-OOD** | **58.93 ± 2.17** | 56.22 ± 4.84 | 47.87 ± 5.46 | **59.13 ± 3.31** |
| **Symmetry** | | | | |
| **ID** | **85.17 ± 1.12** | 85.08 ± 1.25 | **86.07 ± 0.95** | 83.87 ± 0.88 |
| **Near-OOD** | **85.07 ± 1.03** | 83.67 ± 1.33 | 82.17 ± 1.05 | **84.53 ± 0.87** |
| **Far-OOD** | **83.67 ± 0.73** | **82.50 ± 1.00** | 76.00 ± 2.74 | 80.13 ± 1.86 |
| **Spectral** | | | | |
| **ID** | **0.0503 ± 0.0030** | **0.0454 ± 0.0017** | 0.0627 ± 0.0057 | 0.0538 ± 0.0020 |
| **Near-OOD** | **0.1330 ± 0.0111** | 0.1503 ± 0.0149 | 0.1569 ± 0.0078 | **0.1457 ± 0.0139** |
| **Far-OOD** | **0.2090 ± 0.0141** | 0.2325 ± 0.0127 | 0.2296 ± 0.0083 | **0.2194 ± 0.0184** |
| **Bridge** | | | | |
| **ID** | **1.1834 ± 0.1239** | **1.1461 ± 0.0842** | 1.2615 ± 0.0825 | 1.2484 ± 0.0609 |
| **Near-OOD** | **2.5184 ± 0.2428** | 2.8263 ± 0.1644 | 2.8143 ± 0.1015 | **2.3876 ± 0.1618** |
| **Far-OOD** | **6.1317 ± 0.2053** | 6.7032 ± 0.1245 | 6.8447 ± 0.1495 | **5.8798 ± 0.2176** |

# J Preliminary Experiments and Implementation

## J.1 Implementation Details of Preliminary Experiments

We implemented eight neural architectures adapted to graph data. For GNNs, we used GCN, GIN, GAT, and GPS, with 2–5 layers, $128$ hidden units, ReLU activation, dropout with $0.5$, and global mean pooling. GIN and GPS included MLPs in each block, GAT used multi-head attention, and GPS combined local GIN aggregation with global attention. For vision models (ResNet-50, ViT-B/16, Swin-Tiny, ConvNeXtV2-Tiny), graphs were converted to 2D image representations (e.g., adjacency layouts or distance matrices). Models were initialized with ImageNet-1K weights, classification heads replaced by MLPs, and inputs resized to 224×224. All dataset splits were generated using seed 0, with experiments conducted across five random seeds $\in [0, 1, 2, 3, 4]$ for robust evaluation. Models were trained using Adam optimizer (learning rate $5e-6$, weight decay $1e-4$) with batch size 64 and early stopping. For early stopping, the patience settings are set as: 30 epochs for all GNN models, and for vision models, 5 epochs on PROTEINS and 15 epochs on all other datasets.

## J.2 Training dynamics and confidence

Figures 10–13 present training dynamics across four datasets, showing model differences in convergence and generalization. Figures 14 and 15 display confidence distributions for two representative datasets, one from the biological domain (PROTEINS) and one from social networks (IMDB-BINARY), highlighting the contrast between GNNs and vision models in prediction certainty.

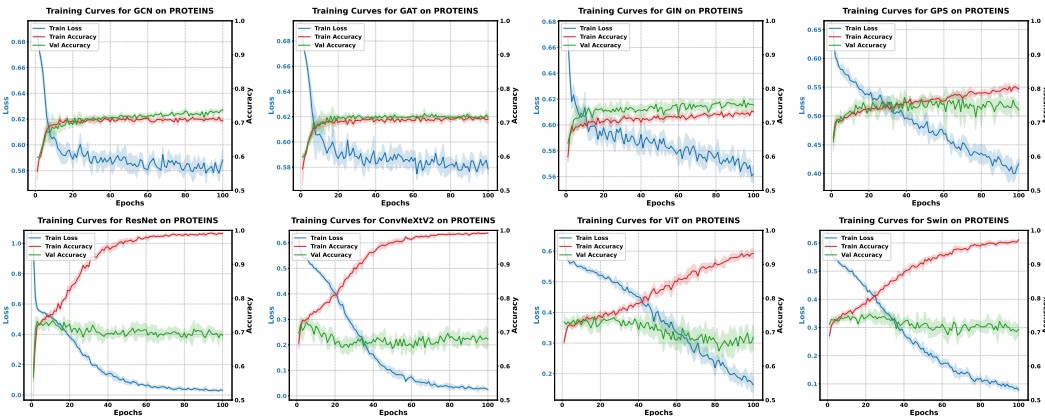

Figure 10: Training dynamics across different architectures on PROTEINS datasets. For each model, we plot the training loss (blue), training accuracy (red), and validation accuracy (green) over 100 epochs. The shaded areas represent the standard deviation across multiple runs.

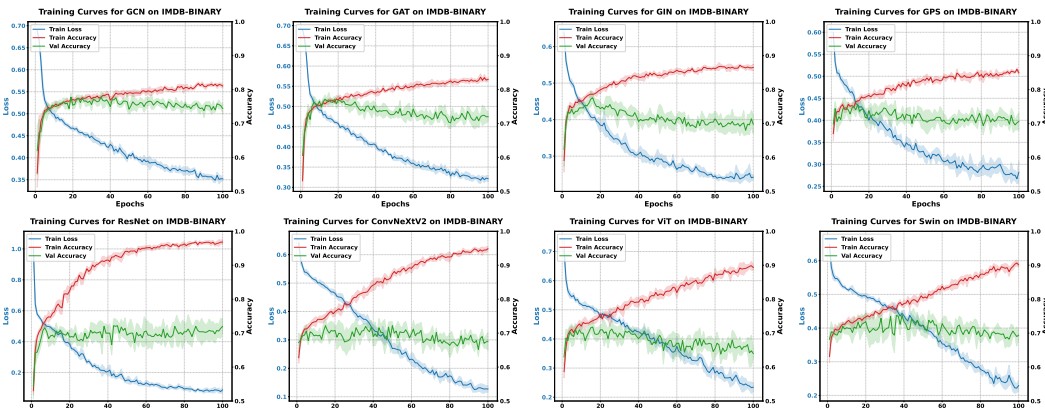

Figure 11: Training dynamics across different architectures on IMDB-BINARY datasets. For each model, we plot the training loss (blue), training accuracy (red), and validation accuracy (green) over 100 epochs. The shaded areas represent the standard deviation across multiple runs.

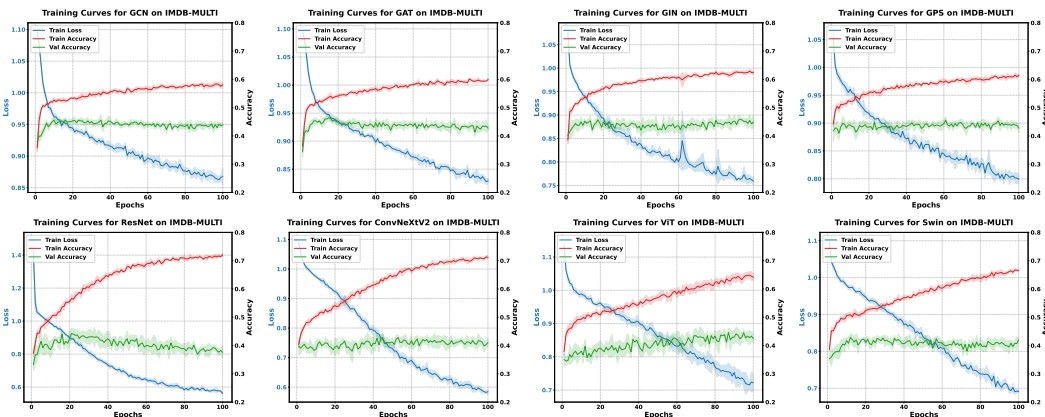

Figure 12: Training dynamics across different architectures on IMDB-MULTI and enzymes datasets. For each model, we plot the training loss (blue), training accuracy (red), and validation accuracy (green) over 100 epochs. The shaded areas represent the standard deviation across multiple runs.

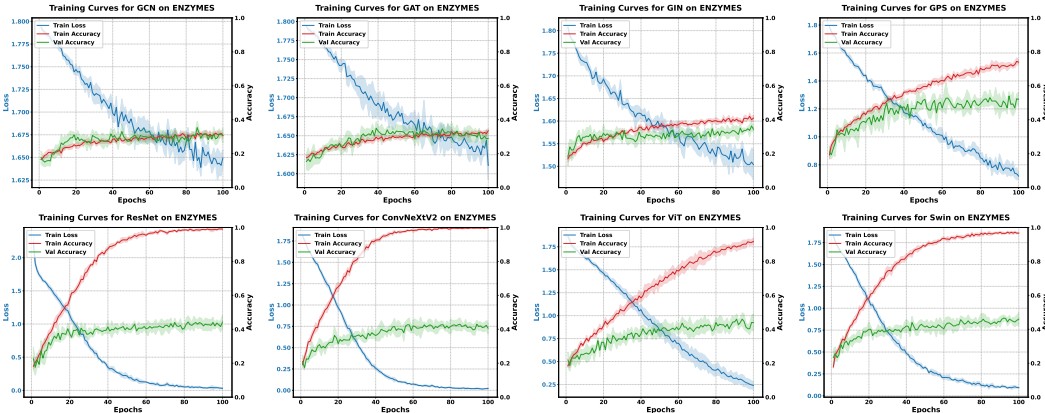

Figure 13: Training dynamics across different architectures on ENZYMES dataset. For each model, we plot the training loss (blue), training accuracy (red), and validation accuracy (green) over 100 epochs. The shaded areas represent the standard deviation across multiple runs.

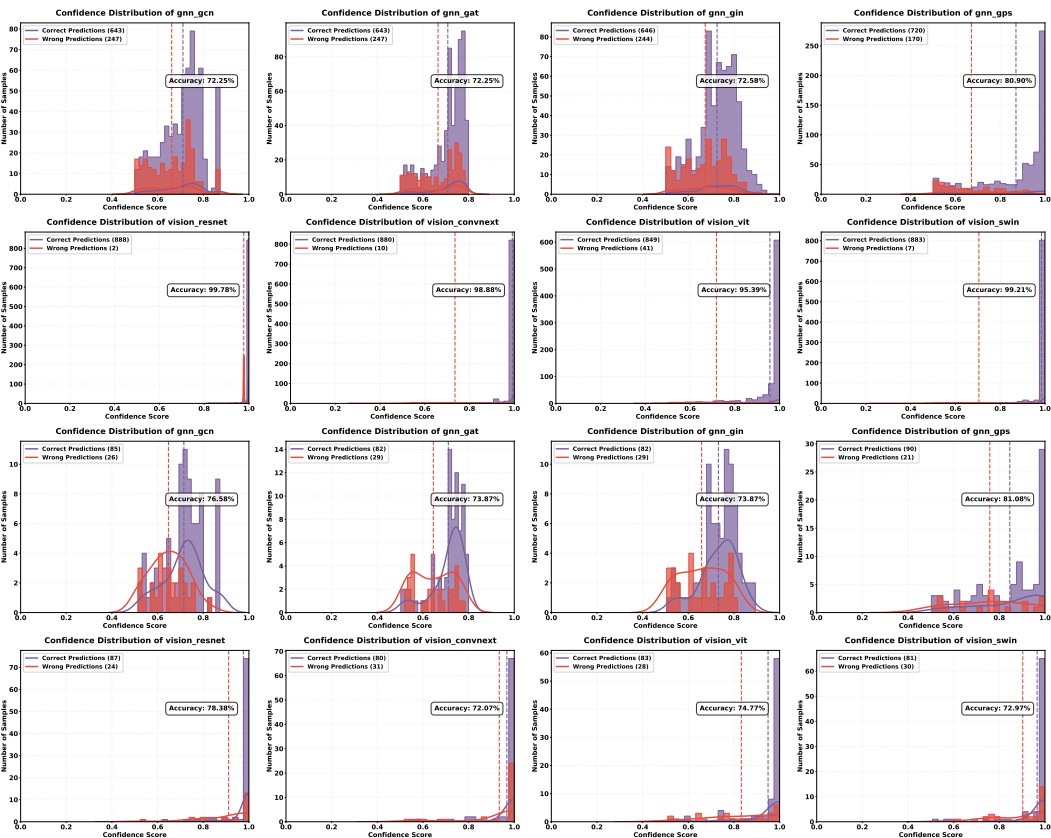

Figure 14: Confidence distribution across different model architectures on the **PROTEINS** dataset. The first two rows show results from the **training set**, while the last two rows present the **test set**. Vision models demonstrate a strong tendency toward high-confidence predictions (0.8-1.0) in both splits, while traditional GNNs typically make lower-confidence predictions. The GPS model, featuring global message passing, uniquely exhibits high-confidence predictions among GNN variants.

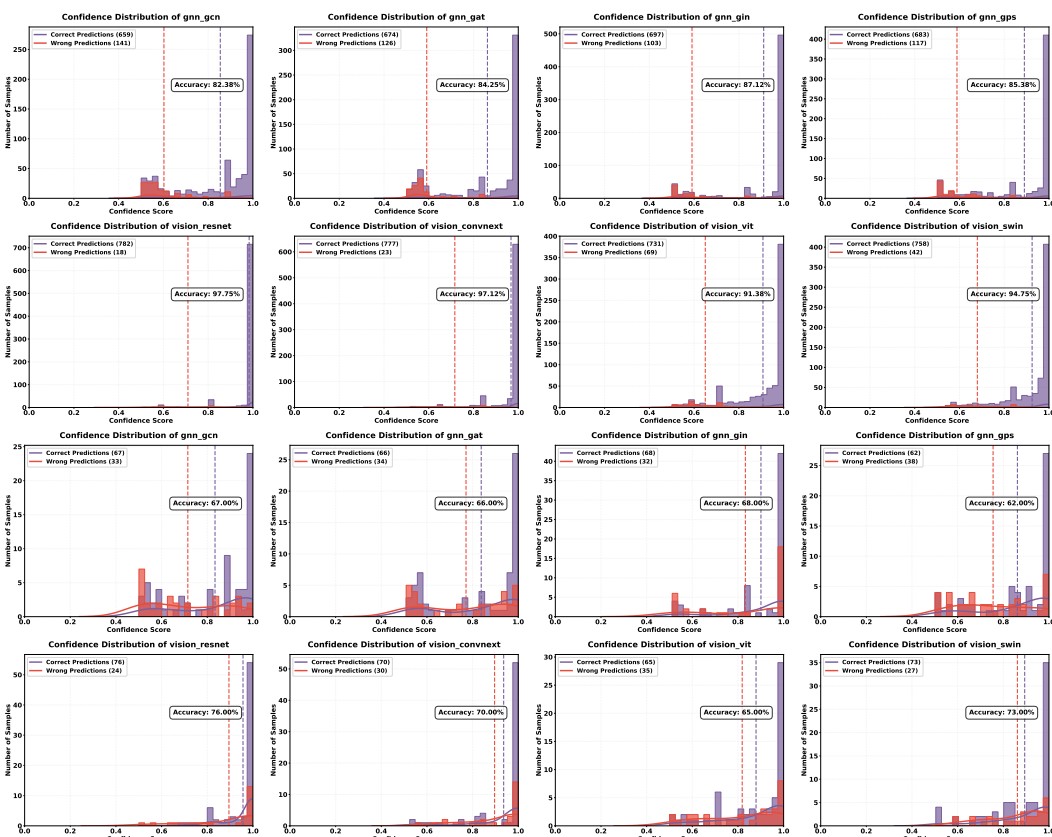

Figure 15: Confidence distribution across different model architectures on the **IMDB-BINARY** dataset. The first two rows show results from the **training set**, while the last two rows present the **test set**.

## J.3 Case Studies

To understand how different models make decisions in graph classification, we explore interpretability methods for both GCN and ConvNeXtV2 as examples. For GCN, we used the GNNExplainer to identify important substructures within input graphs. This approach optimizes masks over edges to highlight influential connections for classification decisions, with visualizations created using `NetworkX` to display node and edge importance. Results across different model depths were compared side by side. For ConvNeXtV2, we applied a Grad-CAM-based approach by registering hooks on target layers to extract activation maps and gradients. The resulting class-specific heatmaps were overlaid on the input images and uniformly displayed to highlight attention differences between models. Figures 16–19 illustrate additional explanation results for GNNs (via GNNExplainer) and vision models (via Grad-CAM) in the first three samples in the testing set of four datasets.

To delve deeper into how these models utilize underlying graph structures for prediction, especially in the presence of known important features, we highlight a specific case study on the ENZYMES dataset. Amongst the datasets considered in our study, ENZYMES offers a unique opportunity for this detailed interpretability analysis, as prior work by [12] has already identified and characterized **discriminative pattern** for it. This pre-existing knowledge allows us to assess alignment with established important features. Figure 16 presents this analysis:

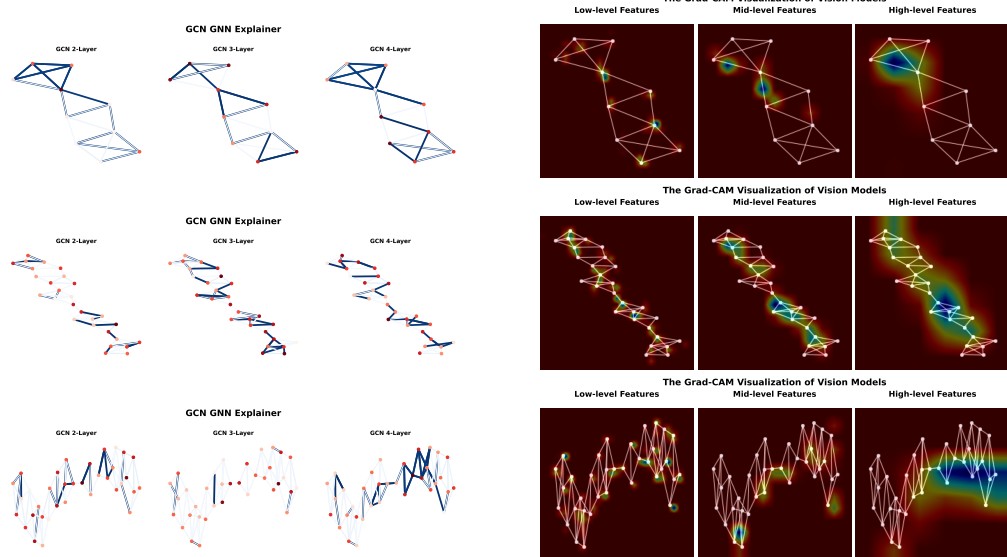

Figure 16: Comparison of GNN explainer (left) and Vision Model Grad-CAM (right) on ENZYMES dataset. The discriminative pattern, defined in [12] as *a square with two diagonal connections* that appears in >90% of graphs within one class but <10% in others, is are key feature for classification. In the top row, where two discriminative patterns exist (one at each end of the graph), while both approaches identify these patterns, the Vision model's Grad-CAM shows particularly sharp focus on them at high-level features, compared to the GNNExplainer's more uniform attribution. In the middle row, the Vision model effectively highlights multiple discriminative patterns near cut-vertices and cut-edges where the graph structure narrows. In contrast, the GNNExplainer shows scattered attention without emphasizing these critical patterns. The bottom row, containing no pre-defined discriminative patterns, demonstrates the different attention patterns of both approaches on non-characteristic structures. These results suggest that Vision models have learned to effectively leverage these discriminative patterns as reliable shortcuts for classification, while GNNExplainers maintain relatively uniform attention distributions.

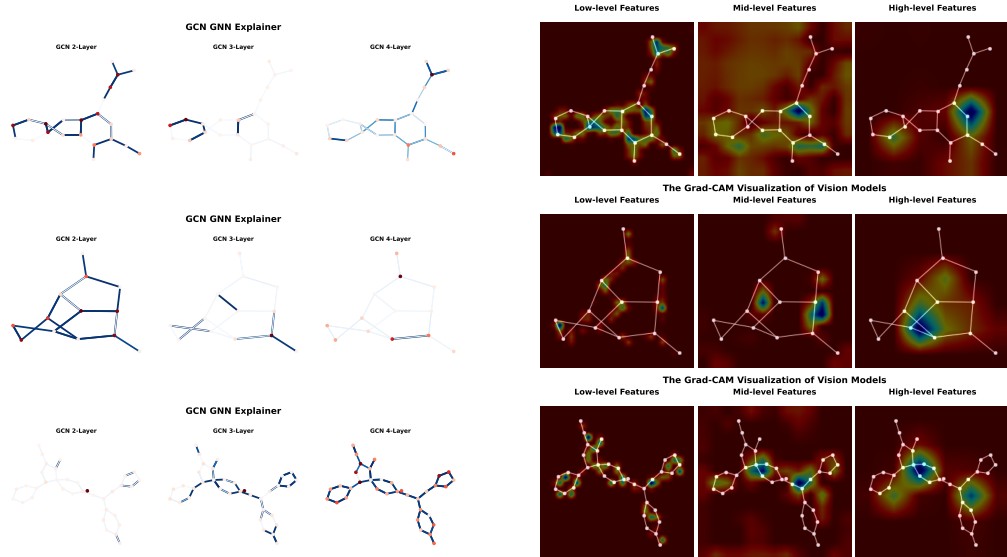

Figure 17: Case Studies for NCI1 dataset.

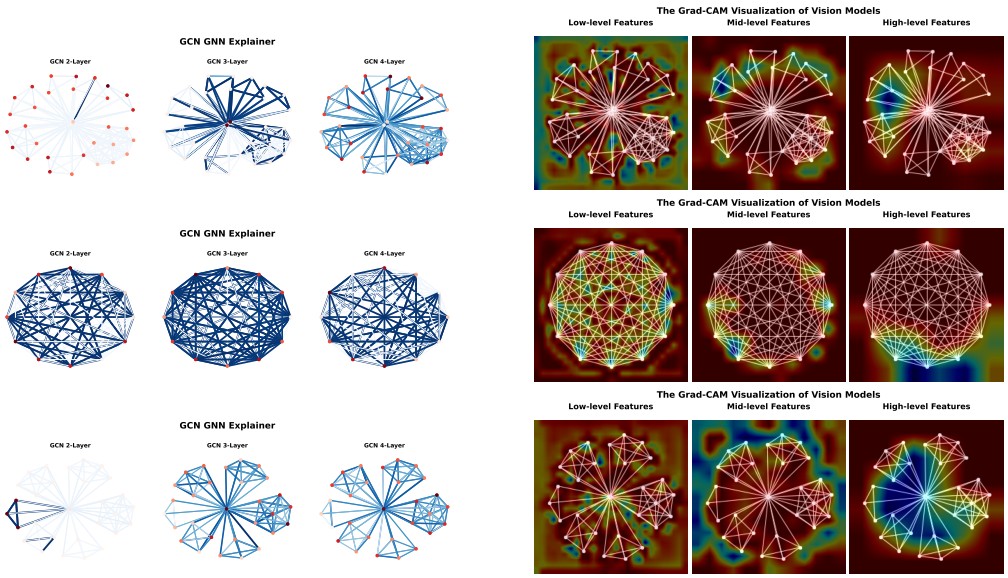

Figure 18: Case Studies for IMDB-BINARY dataset.

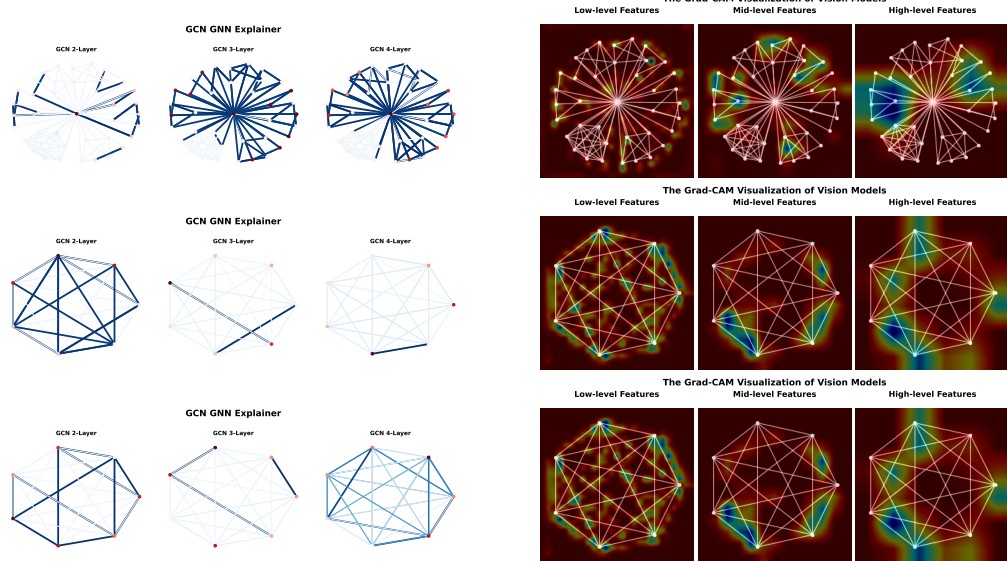

Figure 19: Case Studies for IMDB-MULTI dataset.

## J.4 Heatmap

In this section, we present detailed prediction overlap analysis for all evaluated models across five benchmark datasets. Figures 20–24 illustrate the prediction overlap patterns between GNN models of varying depths (1-6 layers) and vision-based models. Consistent with our main findings, the results show high intra-family similarity among GNNs regardless of layer depth, while maintaining distinctly different prediction patterns compared to vision models.

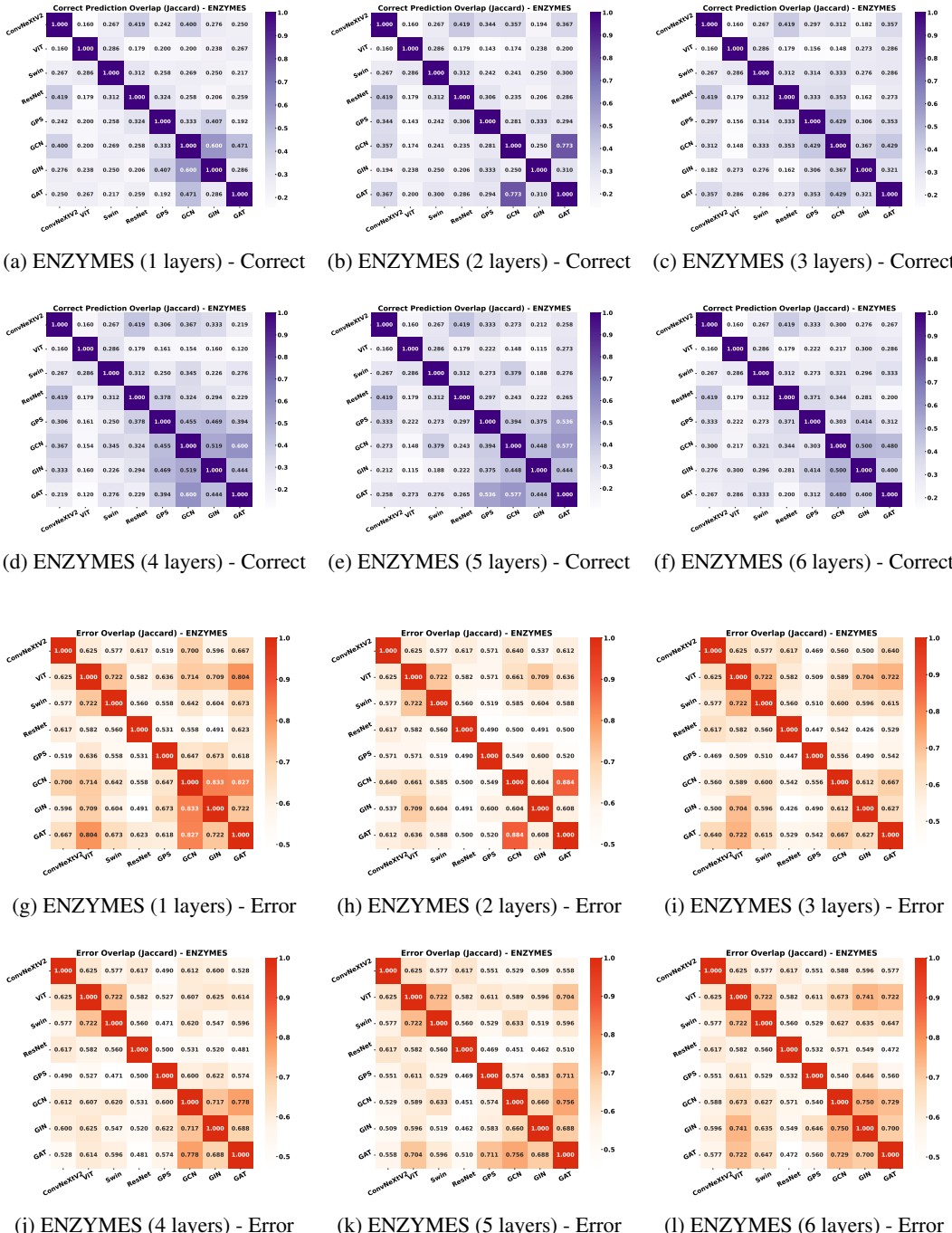

Figure 20: Prediction overlap patterns for ENZYMES dataset with varying GNN layer depths. Top row shows correct prediction overlap, while bottom row shows error overlap patterns across different layer configurations.

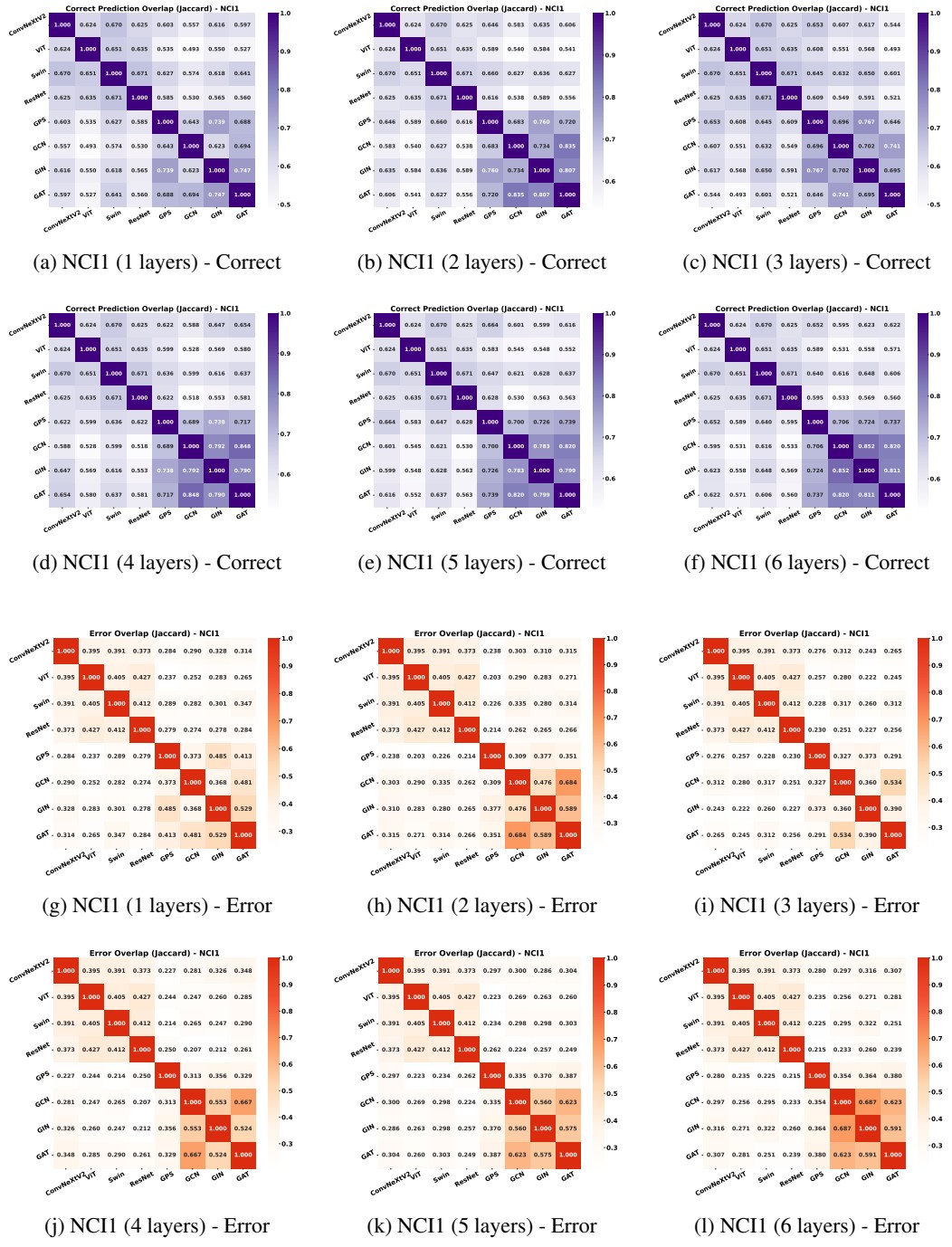

Figure 21: Prediction overlap patterns for NCI1 dataset with varying GNN layer depths. Top row shows correct prediction overlap, while bottom row shows error overlap patterns across different layer configurations.

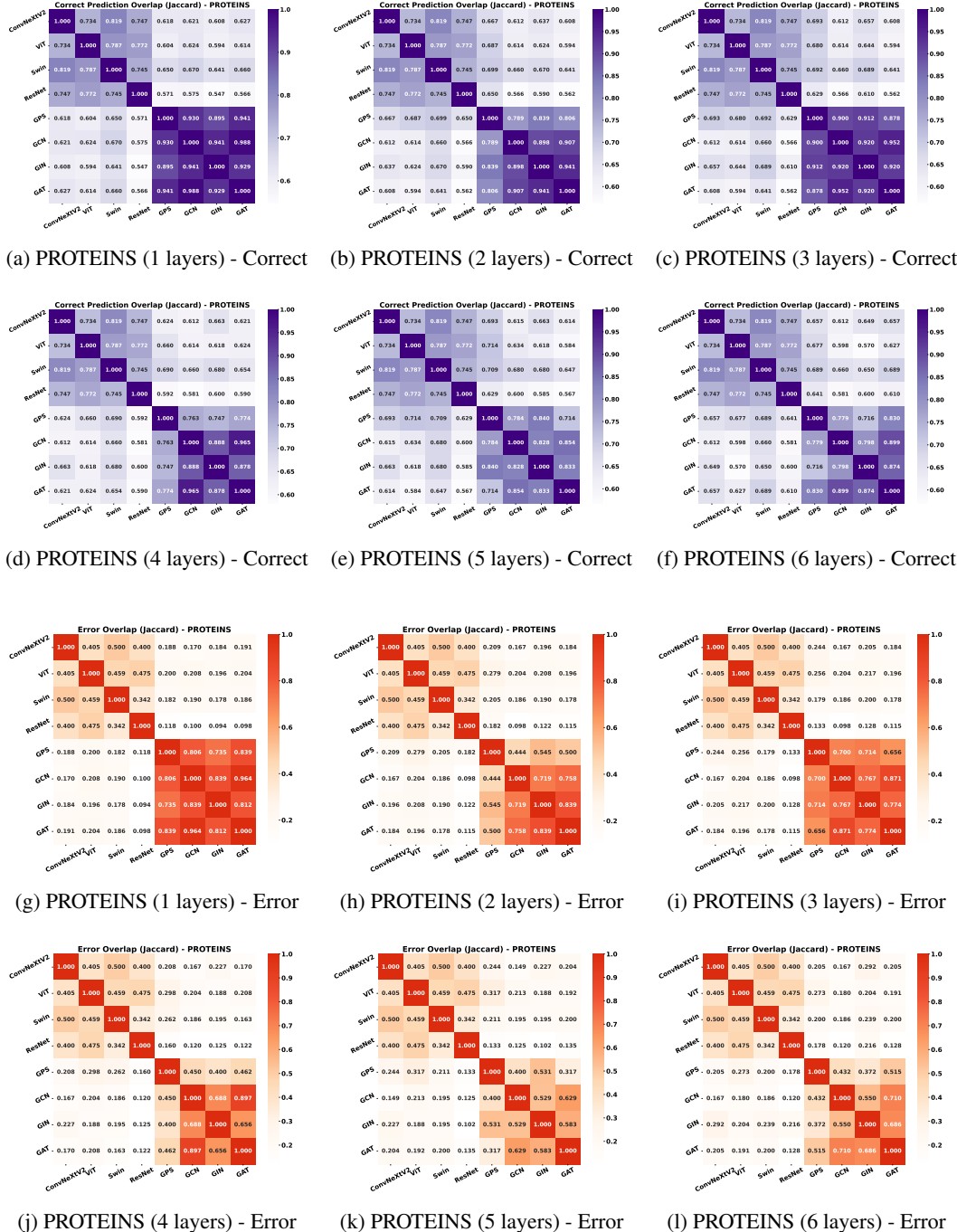

(a) PROTEINS (1 layers) - Correct     (b) PROTEINS (2 layers) - Correct     (c) PROTEINS (3 layers) - Correct

(d) PROTEINS (4 layers) - Correct     (e) PROTEINS (5 layers) - Correct     (f) PROTEINS (6 layers) - Correct

(g) PROTEINS (1 layers) - Error     (h) PROTEINS (2 layers) - Error     (i) PROTEINS (3 layers) - Error

(j) PROTEINS (4 layers) - Error     (k) PROTEINS (5 layers) - Error     (l) PROTEINS (6 layers) - Error

Figure 22: Prediction overlap patterns for PROTEINS dataset with varying GNN layer depths. Top row shows correct prediction overlap, while bottom row shows error overlap patterns across different layer configurations.

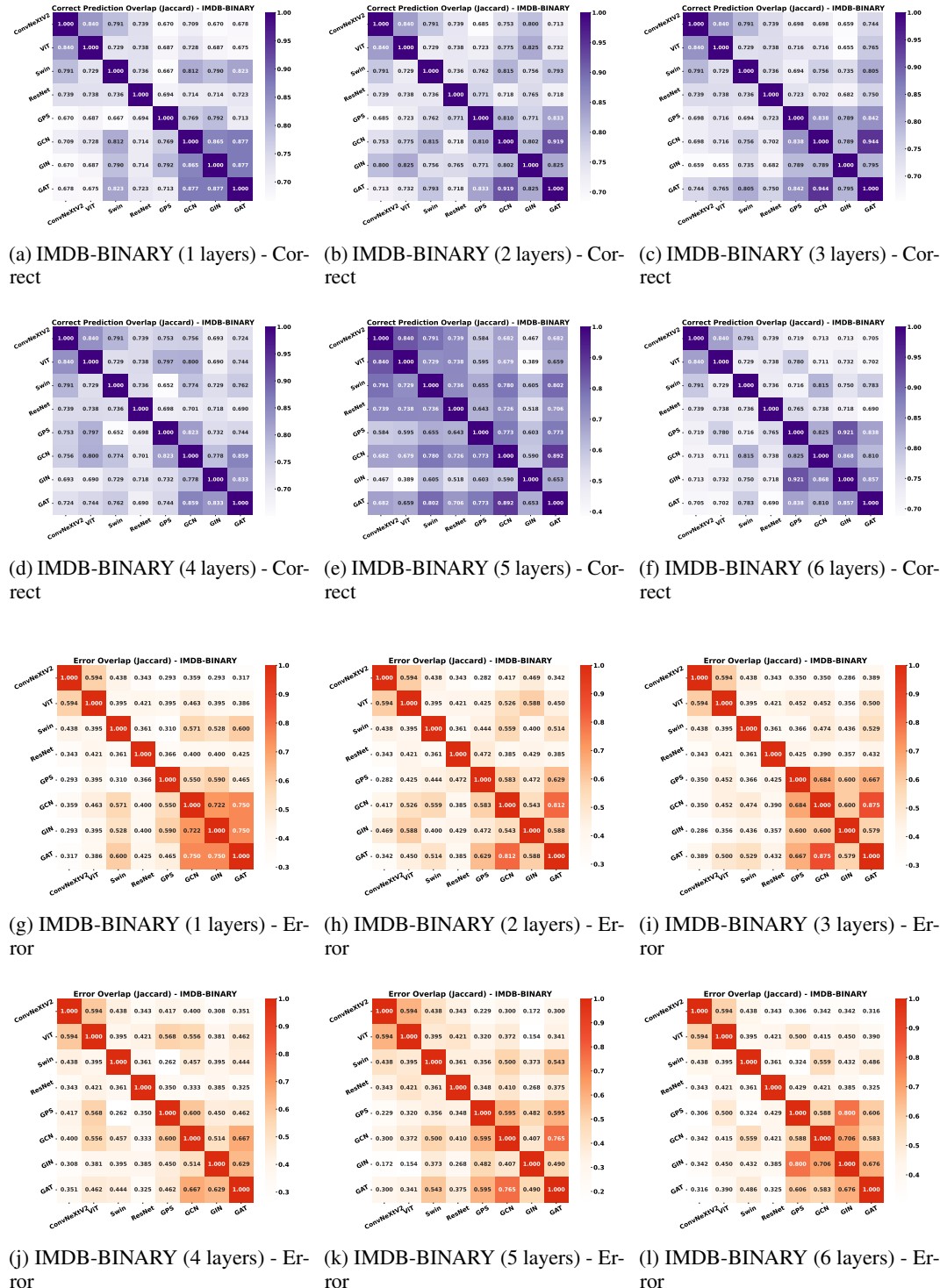

Figure 23: Prediction overlap patterns for IMDB-BINARY dataset with varying GNN layer depths. Top row shows correct prediction overlap, while bottom row shows error overlap patterns across different layer configurations.

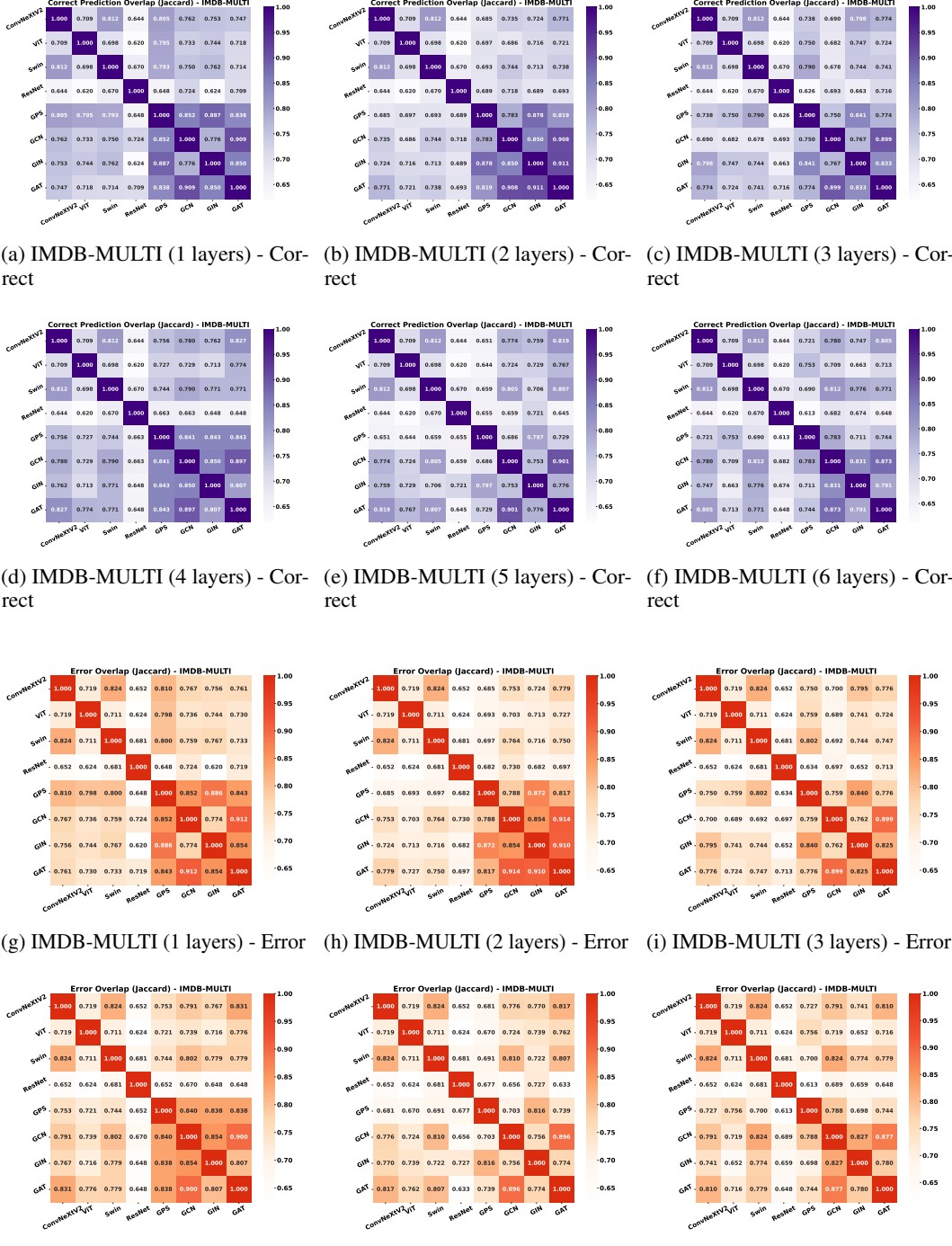

(a) IMDB-MULTI (1 layers) - Correct

(b) IMDB-MULTI (2 layers) - Correct

(c) IMDB-MULTI (3 layers) - Correct

(d) IMDB-MULTI (4 layers) - Correct

(e) IMDB-MULTI (5 layers) - Correct

(f) IMDB-MULTI (6 layers) - Correct

(g) IMDB-MULTI (1 layers) - Error

(h) IMDB-MULTI (2 layers) - Error

(i) IMDB-MULTI (3 layers) - Error

(j) IMDB-MULTI (4 layers) - Error

(k) IMDB-MULTI (5 layers) - Error

(l) IMDB-MULTI (6 layers) - Error

Figure 24: Prediction overlap patterns for IMDB-MULTI dataset with varying GNN layer depths. Top row shows correct prediction overlap, while bottom row shows error overlap patterns across different layer configurations.

