# OpenReview forum: "The Underappreciated Power of Vision Models for Graph Structural Understanding"
_NeurIPS.cc/2025/Conference — NeurIPS 2025 poster_

### Official Review · Reviewer_ofCS · 2025-06-15

**Clarity:** 4
**Significance:** 3
**Originality:** 3
**Rating:** 5
**Confidence:** 5

**Summary:**

This paper explores the use of visual machine learning models, e.g., ResNet, on graph-based applications, addressing the fundamental question of whether existing state-of-the-art graph neural network (GNN) models fully capture global graph properties. To this end, the paper follows a simple strategy: Convert graph structures into visualizations following standard algorithms, and train vision models on the same tasks. Unlike graph models, vision models lose graph structural inductive bias and access to node or graph-level features. Nonetheless, the paper shows that vision models can achieve remarkable performance, often comparable with GNNs, despite these disadvantages. The paper then justifiably asserts that existing benchmarks in the graph learning space conflate global structure learning with low-level feature learning, which effectively offer an alternative means of achieving good performance while ignoring the higher-level structures and patterns that humans intuitively perceive from visualizations. To confirm this point, the paper also reports a comparative study on several graph learning benchmarks, highlighting that GNNs and vision models' predictions differ substantially.

To better study the performance of graph learning models towards structural understanding, the paper proposes a carefully designed set of structure-oriented tasks, namely graph structure classification, symmetry detection, bridge counting, and spectral gap regression, and builds the benchmarks using a series of graph generation and perturbation methods. The paper then conducts an empirical study, highlighting the improved robustness and generalization of vision models, the importance of feature initialization in GNNs as a mitigator for the lack of global feature representations, and highlighting that visual representation choices, i.e., how graphs are converted into images, are critical for obtaining strong vision model performance.

**Questions:**

No questions. Please address suggestions in previous section.

**Ethical Concerns:**

["NO or VERY MINOR ethics concerns only"]

**Final Justification:**

I remain positive about this paper's contributions.

**Quality:**

4

**Strengths And Weaknesses:**

Strengths:
- Very foundational work, which is well-justified, rigorously presented, and methodically executed.
- Addresses a very intuitive question in the graph learning space, and proposes a benchmark that could prove very valuable for the field.
- The discussion on vision models and layout algorithms is insightful, and paves the way for further research on how to inject global structure into state-of-the-art graph representation learning approaches.
- The comparative analysis of vision and graph models is comprehensive and offers a new perspective that can help guide future model design.

Weaknesses:
- The paper proposes a benchmark for structure understanding that is well-justified, but falls short of providing any model-oriented solutions to bridge the gap between vision and graph models. Naturally, this is a research direction in its own right, but I feel that this work would be stronger with experiments that add further nuance. Examples include feature representation through images (e.g., discrete labels as colors), better validating general structural understanding through robustness studies (corrupting graphs and images through e.g, blur / noisy edges and measuring performance change).

- [Minor] The paper makes claims about the conflation of features and structures as means to achieve high-level performance on a series of established benchmarks. I believe the paper would be stronger if it provides results on these benchmarks showcasing the drop-off when features are discarded.

Overall, I think this paper sets up a promising direction for further investigation, which can stimulate research into alternative means for encoding structural graph properties. While there are clear avenues to improve the work (e.g., model proposals for combining features and global visual structure, more studies), I still believe the dataset contribution and insights provided in this work merit publication as-is.

---

> ### Author Rebuttal · Authors · 2025-07-30
>
> We sincerely thank you for your encouraging review and thoughtful suggestions. We have carefully considered each of your points and provide our responses below.
>
> **1. Feature representation through images:**
>
> We appreciate your suggestion about encoding discrete labels as colors. Following your recommendation, we implemented a color encoding scheme where node features (e.g., atom types in molecular graphs) were mapped to distinct colors using hash-based assignment for consistency.
>
> We tested this approach across multiple real-world datasets:
>
> | Model | NCI1 | IMDB-MULTI | ENZYMES | IMDB-BINARY | PROTEINS |
> |-------|------|------------|---------|-------------|----------|
> | **Vision Models (with color encoding)** |
> | ResNet | 69.7±1.0 (+2.0) | 50.1±1.1 (-1.1) | 38.7±1.2 (+2.4) | 72.4±1.0 (+0.2) | 77.8±1.5 (-1.8) |
> | ViT | 67.0±0.9 (+3.5) | 47.1±2.6 (-3.7) | 33.0±4.5 (+5.7) | 72.6±2.7 (+0.8) | 81.6±1.5 (-1.5) |
> | Swin | 73.0±1.0 (+4.0) | 51.2±1.4 (-0.8) | 40.3±5.5 (0.0) | 71.0±1.8 (+2.2) | 82.2±1.1 (+0.4) |
> | ConvNeXt | 73.1±1.3 (+2.9) | 50.4±1.9 (-3.1) | 33.0±5.2 (-8.7) | 67.6±3.3 (-6.2) | 81.1±2.7 (+0.4) |
>
> *Numbers in parentheses indicate change from baseline (without color encoding)*
>
> We observed dataset-specific patterns in these preliminary results:
> - **Consistent improvements** on NCI1 (all models: +2.0 to +4.0%) and IMDB-BINARY (3 out of 4 models positive)
> - **Consistent degradation** on IMDB-MULTI (all models: -0.8 to -3.7%)
> - **Mixed results** on ENZYMES and PROTEINS depending on the model
>
> As preliminary results, these patterns show some promise. The consistent improvements on certain datasets suggest that visual feature encoding has potential for specific graph types. The effectiveness appears to be related to the nature of node features (e.g., discrete atom types in NCI1 vs. degree-based features in IMDB datasets).
>
> Importantly, these experiments used our baseline hyperparameters without any tuning for the color encoding scheme. The dataset-specific patterns, combined with the fact that we achieved improvements without optimization, suggest that more sophisticated visual encoding methods with proper hyperparameter tuning could yield better results. We believe developing effective methods to encode node and edge features for visual graph representation remains a promising direction for future research.
>
>
> **2. Robustness studies (corrupting graphs and images):**
>
>  We greatly appreciate this insightful suggestion. Robustness analysis through systematic corruption of both graph structures and visual representations would indeed provide valuable insights into the different failure modes of GNNs versus vision models. We agree this is an important direction and are currently working on comprehensive robustness experiments involving both graph structural corruptions and visual perturbations. Given the scope of this analysis, we will do our best to provide preliminary results during the discussion phase.
>
> **3. Minor comment on benchmark conflation:**
>
> Thank you for this suggestion. In Section 4.1, we claim that traditional benchmarks "inadvertently couple domain-specific node features with topology, often allowing models to succeed through feature-based shortcuts." You are right that this deserves empirical validation.
> Recent work [1,2] provides compelling evidence: replacing molecular graph structures with fixed Cayley graphs resulted in minimal performance drops or even improvements, suggesting that models may sometimes rely more on node features than structural patterns. We conducted complementary feature ablation experiments to further investigate this phenomenon:
>
> | Model | NCI1 | ENZYMES | PROTEINS |
> |-------|------|---------|----------|
> | **GNN Models (with random features)** |
> | GCN | 52.7±2.7 (-12.7) | 14.3±1.3 (-11.0) | 73.0±3.9 (-1.8) |
> | GAT | 54.7±1.4 (-12.3) | 14.3±4.8 (-12.7) | 75.5±2.4 (+1.4) |
> | GIN | 54.5±3.8 (-15.9) | 15.3±2.2 (-11.0) | 74.6±2.1 (-0.4) |
> | GPS | 54.2±1.4 (-22.1) | 16.3±4.1 (-18.0) | 74.2±1.3 (-1.8) |
>
> *Numbers in parentheses indicate change from baseline (with original features)*
>
> These preliminary results support [1,2]'s findings, showing varying degrees of feature dependence across benchmarks. This evidence strengthens our motivation for GraphAbstract's pure topology focus.
>
> [1] Position: Graph Learning Will Lose Relevance Due To Poor Benchmarks. ICML 2025.
>
>
> [2] Graph Neural Networks Use Graphs When They Shouldn’t. ICML 2024.
>
>
> **4. Future directions for hybrid models :**
>
> We appreciate your suggestion about combining features and global visual structure. While our current work focuses on establishing the benchmark and demonstrating the potential of visual processing, we agree that hybrid approaches represent an exciting future direction. We will add this to our future work discussion.
>
> Thank you again for your encouraging review. Your constructive suggestions have helped us better articulate our contributions and identify promising extensions. We greatly appreciate your thorough feedback.

---

> > ### Comment · Reviewer_ofCS · 2025-08-01
> > **Reviewer Response**
> >
> > I thank the authors for their response.
> >
> > I am pleased that you have found my suggestions helpful, and hope that they prove helpful to your work going forward. The visual feature encoding results are indeed promising. Overall, I am satisfied with the contributions of this paper, and maintain my positive verdict.

---

> > > ### Author Response · Authors · 2025-08-02
> > >
> > > Thank you for your encouraging feedback and sustained support of our work. Your constructive suggestions have been particularly helpful, and we truly appreciate the care you put into reviewing our paper.

---

### Official Review · Reviewer_Aeju · 2025-06-20

**Clarity:** 3
**Significance:** 3
**Originality:** 4
**Rating:** 4
**Confidence:** 3

**Summary:**

This paper discusses the challenges associated with current benchmarks used to evaluate graph AI. Traditional benchmarks, such as node classification and link prediction, entangle node/edge features with the graph's structural properties, making it difficult to assess which aspects contribute most to performance. To address this issue, the authors propose GraphAbstract, a benchmark designed to evaluate models based purely on structural characteristics.

GraphAbstract encompasses tasks such as topology classification, symmetry detection, spectral gap regression, and bridge count regression, with a focus on structural understanding. Moreover, it incorporates evaluation on larger graphs than those seen during training, to assess whether structurally meaningful patterns can be recognized visually at scale.

Using this benchmark, the authors show that vision-based models, which operate on images generated from graph layouts, significantly outperform traditional GNNs. They also demonstrate that incorporating image-style positional encodings into GNNs dramatically improves their performance. These findings suggest that integrating visual and intuitive processing into graph AI could lead to more robust and flexible models for structural understanding.

**Questions:**

1. Can you demonstrate that the graph layouts capture structural features in a fundamentally different way from conventional GNNs? Without a sufficient explanation of this distinction, it is difficult to convincingly argue that GNNs fail to understand structural properties.

2. Many of the structural features included in your benchmark can be effectively addressed using task-specific optimization methods, such as modularity-based approaches. Can you explain what types of problems cannot be solved simply by selecting and applying such specialized methods tailored to each task?

**Ethical Concerns:**

["NO or VERY MINOR ethics concerns only"]

**Final Justification:**

The rebuttal has provided a clearer explanation of how graph visualization can capture features that GNNs cannot. I also understand that, while specific problems can indeed be solved using specialized methods, this work aims to develop a method that can be applied broadly, regardless of the problem type. I hope these explanations will be sufficiently reflected in the final manuscript, and I am inclined to raise my rating by one point.

**Limitations:**

As noted above, the processing pipeline of vision models involves many intertwined factors, including structural interpretation through graph layouts, layout adjustments, image resolution, and more. Because of this complexity, the claim that “GNNs are poor at capturing structural features while vision models are superior” lacks sufficient persuasiveness. A more careful analysis and discussion of these individual components is necessary to convincingly identify the specific limitations of GNNs in understanding structural properties.

**Paper Formatting Concerns:**

No specific concerns regarding paper formatting were identified.

**Quality:**

3

**Strengths And Weaknesses:**

Strengths:
This paper introduces a new benchmark designed to purely evaluate the understanding of structural properties in graphs, which has been difficult to assess with conventional benchmarks. Rather than merely proposing evaluation metrics, the authors go further by detailing how synthetic datasets are generated to faithfully represent the intended structural properties, making the benchmark both practical and conceptually sound. Moreover, the finding that vision models outperform traditional GNNs on this benchmark provides compelling insights into the future direction of graph AI, suggesting the potential power of integrating visual, intuitive reasoning into structural graph understanding.

Weaknesses:
While the proposal of a new benchmark represents a significant contribution, there remain serious concerns regarding the vision-based evaluation used to validate its utility. The most critical issue is the lack of sufficient discussion on the role of graph layout algorithms, which inherently embed structural information into visual representations. Since any layout method captures structural features in some way, failing to analyze how these layouts influence the outcome raises substantial doubts. As a result, evaluating model performance on rendered layouts introduces numerous confounding factors, including layout algorithm biases, resolution effects, and spatial adjustments, making it difficult to isolate the model's structural understanding. Therefore, concluding that vision models inherently understand graph structure seems premature. More careful experimental design and thorough analysis could greatly enhance the credibility and impact of this work.

---

> ### Author Rebuttal · Authors · 2025-07-30
>
> We sincerely thank you for your thoughtful and constructive feedback. Your questions highlight important aspects of our work and provide valuable opportunities to better articulate our approach and findings. We address each point below.
>
> **Weakness & Q1:Can you demonstrate that the graph layouts capture structural features in a fundamentally different way from conventional GNNs?**
>
> Thank you for raising this fundamental question. These two approaches represent different computational pipelines:
> - **GNNs**: Take graph structure (adjacency matrices, edge lists) as input and build understanding through iterative local message passing
> - **Layout + Vision**: Perform global computations upfront (eigendecomposition, energy minimization) to embed graphs in 2D space, then process the resulting geometric patterns
>
> This difference becomes particularly evident when we consider tasks where GNNs have known theoretical limitations. **It's noteworthy that when researchers construct counterexamples for the Weisfeiler-Lehman test and its k-dimensional variants, they almost invariably use graph visualizations to illustrate why these graphs are non-isomorphic despite fooling the WL algorithm [1]**. This reveals an important truth: visual representations can make structural distinctions immediately apparent that formal iterative refinement procedures miss. Another example comes from the TOGL [2]. In their Figure 1, they claim to present datasets "**whose graphs can be easily distinguished by humans.**"  Their NECKLACES dataset shows both classes have the same number of cycles but differ in how these cycles are connected (two individual cycles versus a merged one). According to their theoretical analysis, while humans can immediately see this difference visually, standard message-passing approaches fail to distinguish them. TOGL addresses this by learning sophisticated topological features through persistent homology calculations, but this illustrates our point: what requires complex mathematical machinery for algorithms is claimed to be immediately apparent to human visual perception. Similarly, a set of  GNNs provably cannot identify bridges [3]. However, when graphs are visualized, these structures often manifest as immediately recognizable geometric patterns.
>
> While these examples suggest that visual processing can access structural information differently than message passing, establishing rigorous theoretical proof remains challenging. The field currently lacks formal characterizations of what layout algorithms preserve or how this affects learning. Early work like Eades & Lin [4] connected layouts to graph properties, proving that spring algorithms can display geometric automorphisms. However, over the past decades, the graph visualization field has shifted focus toward human aesthetics rather than developing rigorous theory for machine learning applications. This evolution makes establishing a formal theoretical framework challenging at present. Establishing these theoretical foundations remains essential future work.
>
> In our revised manuscript, we will expand our discussion of both the current limitations and future opportunities. Developing formal connections between geometric properties of layouts and learnability guarantees, and designing layouts optimized for machine perception rather than human aesthetics, could establish a new research area bridging graph visualization and machine learning. We believe this represents an exciting direction for the community to explore.
>
> [1] Wang, Y., & Zhang, M. (2023). An empirical study of realized GNN expressiveness. ICML 2024.
>
> [2] Horn, Max, et al. "Topological graph neural networks."  ICLR 2022.
>
> [3] Zhang, Bohang, et al. "Rethinking the expressive power of GNNs via graph biconnectivity." ICLR 2023
>
> [4] Eades, P., & Lin, X."Spring algorithms and symmetry." Theoretical Computer Science 2000.
>
>
> **Q2 (Limitation 1): Regarding the processing pipeline and confounding factors:**
> Thank you for your thoughtful analysis regarding the potential confounding factors in our layout-based approach. We appreciate the opportunity to clarify our methodology and provide additional experimental evidence.
>
> **Regarding layout algorithm biases, resolution effects, and spatial adjustments:**
>
> We understand your concerns about these factors potentially confounding our results. However, we view these not as limitations but as integral components of the visual processing pipeline that provide valuable insights:
>
> **1. Layout Algorithm Diversity:**
> We deliberately employed multiple layout algorithms (Kamada-Kawai, ForceAtlas2, Spectral, and Circular) with fundamentally different design principles. An important insight is that these algorithms, despite their diversity, are all designed with human visual preferences in mind - they aim to make graph structures interpretable to human vision. Our finding that vision models outperform basic GNNs across these diverse layouts suggests they successfully exploit this common foundation of visual encoding principles that make graphs comprehensible to human perception, regardless of the specific algorithm used.
>
> As discussed in Section 4.4 and Appendix F, our experiments across multiple layouts serve to provide an initial understanding of which approaches suit different tasks:
> - Spectral layouts excel at symmetry detection (8-10% higher accuracy) by naturally manifesting symmetries through geometric arrangements
> - Circular layouts enable straightforward symmetry assessment through edge density patterns
> - Force-directed methods excel at revealing community structures
>
> This diversity helps practitioners understand which layout-task combinations are most effective, providing practical guidance for future applications.
>
> **2. Resolution Robustness:**
> Following your suggestion, we conducted systematic resolution experiments. We selected the challenging symmetry task as a representative example:
>
> | Resolution | ID | Near-OOD | Far-OOD |
> |------------|-------|----------|---------|
> | 448×448 | 80.87 ± 1.80 | 76.27 ± 2.99 | 72.60 ± 4.15 |
> | 224×224 | 84.30 ± 0.36 | 81.07 ± 0.56 | 75.60 ± 2.19 |
> | 128×128 | 84.50 ± 0.46 | 82.73 ± 0.90 | 74.77 ± 1.60 |
> | 64×64 | 86.20 ± 0.69 | 81.03 ± 1.50 | 72.73 ± 2.22 |
>
> *Table 1. ResNet Resolution Study - Kamada-Kawai Layout (Symmetry Task)*
>
>
> | Resolution | ID | Near-OOD | Far-OOD |
> |------------|-------|----------|---------|
> | 448×448 | 93.97 ± 0.96 | 91.53 ± 0.71 | 85.93 ± 0.23 |
> | 224×224 | 93.17 ± 1.15 | 89.47 ± 0.46 | 83.20 ± 0.97 |
> | 128×128 | 93.53 ± 0.69 | 88.30 ± 0.40 | 81.23 ± 0.88 |
> | 64×64 | 91.97 ± 1.27 | 87.17 ± 1.48 | 80.93 ± 1.08 |
>
> *Table 2. ResNet Resolution Study - Spectral Layout (Symmetry Task)*
>
> The approach maintains reasonable performance across different resolutions. Notably, even at 64×64 resolution, both layouts achieve over 70% accuracy on Far-OOD tasks, suggesting practical viability at various computational budgets.
>
> **Spatial Encoding**: Layout algorithms indeed encode structural information spatially. This is precisely the insight we aim to highlight: different computational pathways (iterative message-passing vs. visual pattern recognition after global layout computation) provide complementary strengths for graph understanding.
>
> **Given the exploratory nature of our work, we recognize that we cannot yet provide exhaustive theoretical analysis for all variables in the visual processing pipeline**. The graph visualization field has primarily evolved around human aesthetics rather than machine learning objectives, leaving opportunities for future theoretical development. Our empirical evidence across diverse settings points to several promising research directions:
>  - Developing layout algorithms specifically optimized for machine perception
>  - Establishing theoretical connections between geometric properties and learnability
>  - Creating hybrid approaches that combine visual and message-passing strengths
>
> **Manuscript Revision Plans**: We will enhance our manuscript to better communicate our findings and limitations:
>  - In Section 4.4, we will emphasize that optimal layout selection for different tasks requires both theoretical foundations and empirical exploration, positioning our work as an initial step in this direction
> - In the Introduction, we will more clearly articulate that visual and message-passing approaches offer complementary pathways for graph understanding
>  - We will expand our limitations section to explicitly discuss the confounding factors you raised
>
> These revisions will help present a more balanced view of our contribution while maintaining the value of our empirical findings.
>
> **Q3: Many of the structural features included in your benchmark can be effectively addressed using task-specific optimization methods, such as modularity-based approaches. Can you explain what types of problems cannot be solved simply by selecting and applying such specialized methods tailored to each task?**
>
> You correctly note that specialized algorithms (nauty for automorphism, modularity optimization) solve specific tasks optimally. Our focus is different: we examine whether neural models can develop general structural understanding, a capability important for graph foundation models. While specialized algorithms excel at individual tasks, many real-world applications require reasoning about multiple structural properties simultaneously. A unified model that understands various topological patterns could be more practical than maintaining separate algorithms for each property. Our benchmark measures progress toward this general capability, complementing rather than replacing specialized algorithms.

---

> ### Author Response · Authors · 2025-08-08
> **Thank you for your insightful questions**
>
> Dear Reviewer Aeju,
>
> As we're in the final day of the discussion period, we hope our rebuttal has helped address your questions.
>
> We truly appreciated your essential question about layout algorithms and have provided detailed explanations and additional experiments in response. Similar questions also arose in our **Response Part 2** to `Reviewer 3Uke`, where we further clarified these points. During this discussion phase, our additional experiments and clarifications helped address other reviewers' concerns, and we hope they have similarly addressed yours.
>
>
> We're grateful that reviewers recognized our work as having clear and original contributions (`Reviewer 3Uke`) and being foundational work that addresses an intuitive question in graph learning (`Reviewer ofCS`). These perspectives resonate with your observation about the "compelling insights into the future direction of graph AI" that our work provides.
>
> If you have any remaining questions about our responses, we'd be happy to discuss further. Of course, we understand if time constraints prevent further engagement.
>
> Thank you again for your valuable review.
>
> Best regards,
>
> The Authors

---

### Official Review · Reviewer_3Uke · 2025-06-25

**Clarity:** 3
**Significance:** 2
**Originality:** 3
**Rating:** 4
**Confidence:** 4

**Summary:**

In this work, the authors perform a comparative analysis of graph neural networks (GNNs) and vision models on established graph benchmarks (graph classification only), and a newly introduced benchmark, GraphAbstract, to evaluate graph visual understanding capabilities. GraphAbstract is designed with four tasks to evaluate models’ ability to perceive graph properties in a way that mirrors human visual cognition: topology classification, symmetry classification (using Cayley graphs, bipartite double covers, and Cartesian products of graphs), spectral gap regression and bridge counting.

Experimental results on the established benchmarks demonstrate that vision models achieve performance comparable with GNNs, despite not using graph features.

Results on the novel GraphAbstract show that vision models scale more effectively on out of distribution tasks (graph varying by scale), rely on different layouts depending on the task at hand, and that global position encodings greatly affect GNN performanc

**Questions:**

Questions/Suggestions for actionable improvements that would convince me to update my score:

Q1: Can you include a comprehensive analysis of the costs incurred by a vision model vs a GNN in terms of memory, runtime (per epoch, and total runtime) and compute? Could the vision models be more computationally expensive than GNNs, especially since each graph (in this case relatively small graphs) needs to be converted into an image and hence the comparison is not fully fair?

Q2: Can you motivate why imitating human perception on GNNs to detect topological features is a relevant real world problem beyond the benchmarks introduced here? I find the connection to human intuition interesting but not well motivated and integrated into the work being done here. Additionally, this would help motivate why we would even be interested in situations where graph features are dropped.

Q3: Can you provide a better mathematical grounding for why vision models and GNNs differ so greatly? The architectures of the two are not discussed or compared. CNNs are in fact doing message passing, thus why wouldn’t they suffer from the same locality problem as GNNs?

Q4: There are graph neural networks that are targeting graph structure as well (Boosting the Cycle Counting Power of Graph Neural Networks with I^2-GNNs), such as can you provide an overview of these in related work, and/or include any models specifically designed to excel at these tasks in your benchmarking? Can you include more recent GNN with message passing innovations to show how they would perform?

Q5: Can you share the datasets and the code in a publicly available anonymous github repo?

Q6: Can you clearly show which performance results are within 1std of the best ones?

Nit: The caption for Figure 2 can be improved so that the figure is standalone. Additionally, the captions for Table 1 can be improved to make it clear what each of the columns is illustrating.
Nit: Table 3 should be in the main body since the comparable performance on standard benchmarks is a central part of the first part of your paper. The tables in Figure 1 are symmetric so can be greatly reduced in size to make room.
Nit: Examples of graph structures shown in Appendix should be in the main text, as they illustrate well what GraphAbstract can propose.

**Ethical Concerns:**

["NO or VERY MINOR ethics concerns only"]

**Final Justification:**

The authors convinced me about the premise of the paper by their comparison of CNN/GNN and the added experiments addressed my concerned on the computational differences.

**Limitations:**

The authors include a (short) limitations section, but focus it on comparing their models to human experiments. While interesting, this does not seem extremely relevant to the paper which is about GNNs vs vision models. Authors should discuss limitations in the models they evaluated and scope of their results.

**Paper Formatting Concerns:**

No.

**Quality:**

2

**Strengths And Weaknesses:**

Strengths
S1. The paper is well written with a clear and original contribution. I appreciate the presentation of the methodology which featured a good mix of mathematical formulations and plain English.
S2. The paper presents a comprehensive array of empirical results across a range of different real-world datasets.
S3. The paper introduces a new benchmarking suite, GraphAbstract, made of synthetic graphs designed to separate the challenge of understanding graph topology from learning based on node or edge features, and compare vision models and GNNs.


Weaknesses (see Questions for actionable improvement suggestions to address these)
W1. This paper lacks empirical evidence of the computational efficiency (model complexity, memory, and convergence time) brought on by the method, and how these aspects vary between GNNs and vision models.
W2. The paper does not ground their analysis in real world examples. I am interested in scenarios in which we would want to predict symmetry or bridge counting that extend beyond GraphAbstract, or why we would want the prediction to generalize from small graphs to bigger graphs (OOD experiments). The motivation for why these are desiderata for learning on graphs is not quite clear to me.
W3. CNNs also first compute local features by message passing, while the global features only appear at the last layer. Hence, the argument that vision models capture global features seems debatable from the very start.
W4. The GNNs compared are mostly from before 2020 (precisely: 2016, 2017, 2018 and 2022)  while the vision models are mostly from after 2020 (precisely: 2016, 2021, 2020, 2023). In order to conclude that vision models outperform GNNs, or to claim that injecting “pre-computed global structural information significantly outperforms innovations in message passing architectures alone” it would be more fair to include the latest GNNs with appropriate “innovation”.
W5. The dataset and the code are not publicly available, which is a weakness for a benchmark paper.

This paper would have been better suited for the NeurIPS Benchmarks and Dataset Track: Why not submitting there?

---

> ### Author Rebuttal · Authors · 2025-07-30
>
> We sincerely thank the reviewer for the comprehensive and insightful feedback. Your thoughtful comments have been invaluable in helping us improve the quality and clarity of our manuscript. We address each point below.
>
> **W1 & Q1: Computational cost analysis**
>
> Thank you for this suggestion. We conducted comprehensive experiments on a single A100 GPU across all datasets. Due to space constraints, we present topology classification results as a representative example. The complete analysis will be included in the revised manuscript.
>
> | Model | GPU Memory (MB) | Parameters (M) | Time/Epoch (s) | Time to Best Val Acc (s) |
> |-------|-----------------|----------------|----------------|------------------|
> | CONVNEXT | 22980 | 28.2 | 13.1 | 329.5 |
> | RESNET| 10916 | 25.6 | 5.5 | 120.7 |
> | SWIN | 13515 | 27.8 | 11.5 | 368.9 |
> | VIT | 18510 | 86.1 | 23.1 | 509.9 |
> | GAT+SPE | 326 | 0.1 | 1.2 | 43.2 |
> | GCN+SPE | 325 | 0.1 | 1.2 | 45.1 |
> | GIN+SPE | 327 | 0.1 | 1.1 | 47.0 |
> | GPS+SPE | 376 | 0.5 | 1.4 | 26.9 |
> | | | | | |
> | *Avg. Vision* | 16480 | 41.9 | 13.3 | 332.2 |
> | *Avg. GNN+SPE* | 338 | 0.2 | 1.2 | 40.6 |
> | **Ratio (V/G)** | **48.7×** | **209.6×** | **10.9×** | **8.2×** |
>
> These results help readers understand the computational tradeoffs between model families. While vision models require more resources, their larger capacity and complex architectures also provide opportunities for incorporating diverse graph-specific inductive biases through pre-training or architectural design. The complete cost analysis across all tasks will be included in the revised manuscript.
>
> **W2 & Q2:  Scenarios in which we would want to predict symmetry or bridge counting that extend beyond GraphAbstract, or why we would want the prediction to generalize from small graphs to bigger graphs.**
>
> Thank you for pushing us to clarify the practical relevance.
>
> **"Scenarios in which we would want to predict symmetry or bridge counting":**
>
> These capabilities serve as diagnostic tests for structural understanding, similar to using arithmetic to assess mathematical reasoning. While not always the direct goal, models lacking these basic capabilities face significant limitations in practice:
>
>  * **Symmetry blindness**: In combinatorial optimization, models unable to recognize symmetries cannot naturally prune equivalent solutions, requiring separate preprocessing to perceive symmetry [1]. This lack of intrinsic understanding prevents the development of truly powerful end-to-end neural solvers.
>
> * **Bridge insensitivity**: In retrosynthesis, models without awareness of critical connections cannot autonomously identify strategic disconnection points or stability-critical structures. This forces reliance on hand-crafted rules or extensive feature engineering, limiting their ability to discover novel strategies [2].
>
> **"Why we would want the prediction to generalize from small graphs to bigger graphs":**
>
> Scale generalization tests whether models truly understand structural concepts. Real-world graphs vary dramatically in size, from small molecular fragments with dozens of atoms to large protein complexes with thousands, from local social groups to entire online networks with millions of nodes. A model that cannot recognize fundamental topological properties across these scales hasn't truly learned what defines these structures. This scale-invariant understanding is a hallmark of genuine comprehension: just as humans recognize a chair whether it's a toy or full-sized furniture, a capable graph model should recognize structural patterns regardless of size. Models that fail this test likely rely on superficial pattern matching rather than deep structural understanding.
>
> In summary, our benchmark represents an initial exploration into the vast space of topological capabilities that humans naturally possess. We selected only a small subset of these abilities for early investigation, believing that systematically identifying and measuring these capability gaps provides insights into the current state of graph learning models. Understanding these fundamental capabilities may prove essential as graph learning evolves toward tasks demanding genuine structural reasoning beyond current approaches.
>
> In the revised manuscript, we will:
>  - Add a dedicated section discussing the relationship between fundamental capabilities and practical limitations
>  - Clearly position our work as diagnostic rather than immediately applicable
> - Expand limitations to acknowledge the gap between our benchmark and real-world impact
>
> [1] When GNNs meet symmetry in ILPs: an orbit-based feature augmentation approach. ICLR 2025.
>
> [2] Unbiasing retrosynthesis language models with disconnection prompts. ACS Central Science 2023.
>
> **W3 & Q3: CNNs also perform local operations. Why don't they suffer from the same locality problem as GNNs?**
>
> You make an important point - CNNs do perform local operations that could be viewed as a form of spatial message passing. The key insight is not that CNNs are fundamentally different in operation, but that the combination of layout preprocessing and CNN processing creates a different computational pathway.
>
> When we examine how CNNs succeed at global tasks in computer vision (scene understanding, object detection), it's through hierarchical feature learning - local edges combine into textures, then shapes, then objects. For graphs, the crucial difference is that layout algorithms already perform global computations before the CNN ever sees the image. Spectral layouts use eigenvectors of the graph Laplacian. Force-directed layouts simulate global energy minimization. These global computations get "baked into" the 2D positions.
>
> So when a CNN processes a graph layout, its local operations are processing the results of global computations. A symmetric graph produces a symmetric layout not by chance, but because the layout algorithm preserves this global property in the spatial arrangement. The CNN then detects this geometric symmetry through local filters(We kindly refer to **Weakness & Q1 to Reviewer `Aeju`** for illustrative examples of how this enables visual models to recognize patterns that challenge message-passing approaches.).
>
> This "global-first" approach contrasts with GNNs that must build global understanding from local operations. Interestingly, this explains why positional encodings (PE) help GNNs - they inject pre-computed global information before message passing begins, providing another form of "global-first" mechanism. Our results confirm this insight: PE-enhanced GNNs show significant improvements, though vision models still excel at tasks requiring holistic pattern recognition.
>
> **W4 & Q4: Coverage of recent GNN developments**
>
> Thank you for this important suggestion about coverage of recent developments. You're right that we should include structure-aware GNNs like subgraph GNNs. We tested I²-GNN as a representative of this approach. The results show it significantly outperforms vanilla GNNs and achieves competitive performance with PE-enhanced models:
> | Task | ID | Near-OOD | Far-OOD |
> |------|-------|----------|---------|
> | Topology | 94.67 ± 1.49 | 87.87 ± 4.90 | 63.33 ± 5.83 |
> | Symmetry | 90.80 ± 3.49 | 84.00 ± 2.53 | 83.40 ± 3.56 |
> | Spectral | 0.1044 ± 0.0091 | 0.3315 ± 0.0941 | 0.7893 ± 0.2709 |
> | Bridge | 0.4580 ± 0.1131 | 1.0459 ± 0.1523 | 3.7353 ± 1.0961 |
> *Table 1: I²-GNN Performance*
>
> The results show that I²-GNN achieves strong performance compared to vanilla message-passing GNNs. Your intuition about pre-computed global information proves particularly insightful here. Regarding your questions about whether "vision models outperform GNNs" or "pre-computed global structural information significantly outperforms innovations in message passing architectures alone," our experiments confirm both: comparing with Table 1 in our paper, vision models maintain overall advantages (e.g., 90.33% vs 63.33% on Far-OOD topology), AND pre-computed global information is indeed crucial for these tasks.
>
>
> I²-GNN, PE methods, and vision models all benefit from different forms of global structure awareness. We view these not as competing approaches but as complementary paradigms: GNN innovations focus on identifying and addressing theoretical limitations (e.g., WL-expressivity, substructure counting), while vision approaches explore human-like pattern recognition. Each offers unique advantages for different aspects of graph understanding.
>
>
> In our revision, we will:
> - Add a comprehensive related work section covering subgraph GNNs, graph transformers, and positional encoding methods
> - Include I²-GNN results in our main comparison tables
>
> **W5 & Q5: Code and Dataset Availability**
>
> Our apologies for the confusion. Code and datasets are in the **supplementary materials**. We will add a prominent URL in the abstract and introduction for easier access in the next version.
>
> **Q6: Statistical Significance**
>
> We will clearly mark (bold/underline) all results within 1 standard deviation of the best performance.
>
> **Limitations**
>
> You're absolutely right that our limitations section should focus more on model limitations rather than human experiments. We will revise it to discuss: the computational overhead of vision models, and the lack of theoretical foundation connecting layout algorithms to learning guarantees. Establishing such a theory represents an exciting future direction.
>
> **Why not Benchmark Track**
>
> We chose the main track because our contribution extends beyond the benchmark itself - we provide fundamental insights into how different computational paradigms (message-passing vs. visual pattern recognition) access graph structure. The benchmark serves to validate these insights rather than being the sole contribution.
>
> **Nits**
>
> Thank you for these helpful suggestions. We will move Table 3 to the main body, improve Figure 2 and Table 1 captions for standalone clarity, and include example graphs from the appendix in the main text.

---

> > ### Comment · Reviewer_3Uke · 2025-08-04
> > **Thank you for clarifying most of the concerns. Remaining questions below.**
> >
> > I thank the authors for their thoughtful and very clear rebuttal.
> >
> > Many of my concerns have been addressed. However I have one remaining (big) concern and one comment.
> >
> > **Concern following W1 & Q1: Computational cost analysis.**
> >
> > Thank you so much for providing the computational cost analysis. I find it very insightful and definitely needed for this paper.
> >
> > Since vision models use so much more memory and computing power, I am concerned that the whole comparison between GNN and vision models is unfair. How would GNN perform with a higher memory/compute budget? How would vision models perform with a lower memory/compute budget?
> >
> > If possible, I would like to see the computational cost analysis tables for the other 3 benchmark tasks.
> >
> > **W3 & Q3: CNNs also perform local operations.**
> >
> > I agree with you that the layout operation immediately creates global information that is passed to the CNN. The analogy with the positional encodings (PE) is very good and really helps understanding this point.
> >
> > However, once the graph has become an image, the CNN still processes it locally, patch by patch. Hence, I still do not understand the rationale given by the authors to justify that the CNN will perform better than GNN+PE. In my opinion, the improve of performance comes from the fact that the vision models are significantly bigger than the GNN models.
> >
> > Again, I thank the authors for their rebuttal. As mentioned previously, I am willing to increase my score but I cannot increase it just yet as I still have doubts about the very premise of the paper (that GNN underperform because they only use local information -- I think it's because they are smaller).

---

> > > ### Author Response · Authors · 2025-08-05
> > > **Response Part 1: Computational Analysis and Model Scaling**
> > >
> > > We sincerely thank the reviewer for the thoughtful follow-up and appreciate your willingness to engage with our work.
> > >
> > > **Concern following W1 & Q1: Computational cost analysis.**
> > >
> > > We are happy to provide the detailed computational analysis across all four benchmark tasks as requested. The results confirm that the computational patterns remain highly consistent - vision models require approximately 10-12× more resources than GNNs+PE across all task types. These comprehensive timing results will be included in the revised manuscript to help readers fully understand the computational tradeoffs between model families:
> > >
> > > | Model | Topology Classification |  | Symmetry Classification |  | Bridge Counting |  | Spectral Gap Regression |  |
> > > |-------|------------------------|--|-------------------------|--|-----------------|--|-------------------------|--|
> > > | | Time/Epoch (s) | Time to Best Val Acc (s) | Time/Epoch (s) | Time to Best Val Acc (s) | Time/Epoch (s) | Time to Best Val Acc (s) | Time/Epoch (s) | Time to Best Val Acc (s) |
> > > | **CONVNEXT** | 13.1 | 329.5 | 9.4 | 245.2 | 11.2 | 573.6 | 13.2 | 717.7 |
> > > | **RESNET** | 5.5 | 120.7 | 4.2 | 76.5 | 4.8 | 127.5 | 5.6 | 312.0 |
> > > | **SWIN** | 11.5 | 368.9 | 8.2 | 123.1 | 9.8 | 227.1 | 11.6 | 348.2 |
> > > | **VIT** | 23.1 | 509.9 | 15.8 | 127.0 | 19.5 | 858.6 | 23.2 | 1139.7 |
> > > | **GAT+SPE** | 1.2 | 43.2 | 0.9 | 16.3 | 1.1 | 2.5 | 1.2 | 107.7 |
> > > | **GCN+SPE** | 1.2 | 45.1 | 0.8 | 5.3 | 1.0 | 93.3 | 1.2 | 35.1 |
> > > | **GIN+SPE** | 1.1 | 47.0 | 0.8 | 6.1 | 1.1 | 56.0 | 1.2 | 30.6 |
> > > | **GPS+SPE** | 1.4 | 26.9 | 1.0 | 22.0 | 1.2 | 34.9 | 1.5 | 32.7 |
> > > | | | | | | | | | |
> > > | *Avg. Vision* | 13.3 | 332.2 | 9.4 | 142.9 | 11.3 | 446.7 | 13.4 | 629.4 |
> > > | *Avg. GNN+SPE* | 1.2 | 40.6 | 0.9 | 12.4 | 1.1 | 46.7 | 1.3 | 51.5 |
> > > | **Ratio (V/G)** | **10.9×** | **8.2×** | **10.7×** | **11.5×** | **10.3×** | **9.6×** | **10.5×** | **12.2×** |
> > >
> > >
> > > **Regarding Computational Cost and Model Size:**
> > >
> > > To address your important point about model size differences, we conducted scaling experiments by testing two larger versions of GPS+SPE.  Our original implementation used a hidden dimension of 128, consistent with common GNN practice. We then expanded hidden dimensions to 1024 and 2048 with 8 attention heads and 5 layers, resulting in models with **53.2M** and **212.2M** parameters respectively.
> > > This matches and even exceeds the parameter budget of vision models (ResNet: **25.6M** parameters) to ensure a fair comparison.
> > >
> > > If the performance gap were simply due to insufficient model capacity rather than fundamental architectural differences, one would expect substantial improvements from this scaling.
> > > However, here are the actual results (with 5 seeds):
> > >
> > >
> > > | Task | Model | ID | Near-OOD | Far-OOD |
> > > |------|-------|:---:|:---:|:---:|
> > > | **Topology** | Baseline GPS+SPE | 84.80 ± 13.75 | 84.07 ± 16.04 | 72.20 ± 14.51 |
> > > | | GPS+SPE (53.2M) | 87.73 ± 6.87 | 70.13 ± 13.03 | 40.53 ± 11.06 |
> > > | | GPS+SPE (212.2M) | 82.53 ± 5.58 | 66.20 ± 11.42 | 34.80 ± 8.16 |
> > > | | ResNet (25.6M) | 95.87 ± 0.62 | 96.27 ± 1.02 | 87.40 ± 3.33 |
> > > | **Symmetry** | Baseline GPS+SPE | 71.97 ± 1.65 | 70.67 ± 1.23 | 67.70 ± 1.37 |
> > > | | GPS+SPE (53.2M) | 65.83 ± 2.21 | 65.83 ± 2.83 | 67.63 ± 3.94 |
> > > | | GPS+SPE (212.2M) | 56.10 ± 4.47 | 55.67 ± 5.75 | 54.97 ± 4.53 |
> > > | | ResNet (25.6M) | 93.47 ± 0.66 | 88.83 ± 0.64 | 84.20 ± 0.39 |
> > > | **Spectral Gap** | Baseline GPS+SPE | 0.0681 ± 0.0298 | 0.1537 ± 0.0839 | 0.6716 ± 0.2709 |
> > > | | GPS+SPE (53.2M) | 0.1483 ± 0.0210 | 0.1901 ± 0.0167 | 0.7497 ± 0.4243 |
> > > | | GPS+SPE (212.2M) | 0.1214 ± 0.0365 | 0.2125 ± 0.0373 | 0.9101 ± 0.5625 |
> > > | | ResNet (25.6M) | 0.0335 ± 0.0021 | 0.0600 ± 0.0063 | 0.1102 ± 0.0100 |
> > > | **Bridge Count** | Baseline GPS+SPE | 0.6402 ± 0.1753 | 1.4666 ± 0.0713 | 3.8021 ± 1.0492 |
> > > | | GPS+SPE (53.2M) | 1.4502 ± 0.2315 | 3.0053 ± 0.5616 | 5.6101 ± 0.8129 |
> > > | | GPS+SPE (212.2M) | 2.1581 ± 0.8106 | 3.3334 ± 0.7540 | 5.7994 ± 1.1003 |
> > > | | ResNet (25.6M) | 0.7771 ± 0.1095 | 1.6356 ± 0.1643 | 3.6814 ± 0.1217 |
> > >
> > > The scaled models showed decreased performance compared to the baseline, with the 212.2M parameter model experiencing the most significant decline. This suggests that simply scaling GNN parameters does not address the performance gap we observe.

---

> > > ### Author Response · Authors · 2025-08-07
> > > **Thank you for the valuable discussion**
> > >
> > > Dear Reviewer 3Uke,
> > >
> > > As the discussion period draws to a close, we hope our additional experiments and explanations have been helpful in addressing your concerns. We greatly appreciate the thoughtful questions you raised, which motivated us to conduct these scaling experiments and clarify the architectural differences. Should you have any remaining questions, we would be happy to provide further clarification.
> > >
> > > If these clarifications have helped address your concerns, we would appreciate if you could reevaluate our work's contributions.
> > >
> > > Thank you again for your valuable engagement with our work.
> > >
> > > Best regards,
> > >
> > > The Authors

---

> > > > ### Comment · Reviewer_3Uke · 2025-08-07
> > > > **Thank you for addressing my concerns!**
> > > >
> > > > I sincerely thank the authors for their detailed and extensive answer, which addresses my concerns.
> > > >
> > > > I will raise my score to 4.

---

> > > > > ### Author Response · Authors · 2025-08-08
> > > > > **Grateful for your thoughtful review and support!**
> > > > >
> > > > > We sincerely thank you for your thoughtful engagement and positive evaluation. Your insightful questions and comments have truly strengthened our work. We greatly appreciate your constructive feedback and support.

---

> ### Author Response · Authors · 2025-08-05
> **Response Part 2**
>
> **W3 & Q3: CNNs also perform local operations**
>
> You correctly note that CNNs also use local operations. Our scaling experiments suggest that the key difference lies not in model size but in what problem each architecture is solving.
>
> Once graphs are rendered as images via layout algorithms, the task transforms from graph analysis to visual pattern recognition. This is a domain where CNNs inherently excel at capturing global patterns, whether classifying complex objects in ImageNet, understanding entire scene layouts in semantic segmentation, or recognizing faces across different poses and conditions. Just as humans can immediately perceive global patterns in visualized graphs (seeing symmetries, clusters, connectivity strength, and critical bridges at a glance), CNNs process these geometric patterns through their hierarchical architecture. From a computer vision perspective, detecting symmetric patterns, identifying clustered regions, or recognizing bottleneck structures are basic geometric pattern recognition tasks that CNNs handle effectively across many applications.
>
> **Supporting observations** from our response to Reviewer `Aeju`: As we detailed in their **Weakness & Q1**, the difference between these approaches becomes evident when considering tasks with known GNN limitations:
> - When researchers construct counterexamples for the WL test, they invariably use visualizations to show why graphs are non-isomorphic, because visual representations make structural distinctions immediately apparent that iterative refinement procedures miss [1]
> - Horn et al. [2] explicitly present datasets "whose graphs can be easily distinguished by humans" visually, where structural differences require complex mathematical machinery (persistent homology) for GNNs but are immediately visible to humans
> - Bridges, which GNNs provably cannot identify [3], often manifest as obvious visual bottlenecks in layouts
>
> These examples from our community's own practices suggest that layout algorithms transform the problem: from iterative message passing through graph edges to pattern recognition in geometric space, where structural properties become directly observable patterns.
> We hope this perspective helps explain our findings. While both architectures use local operations, they access and process structural information through different pathways.
>
> We greatly appreciate your thoughtful feedback which motivated these important experiments. We hope this additional analysis addresses your concern about model size being the primary factor and is helpful for your evaluation. Thank you again for your valuable insights that have strengthened our work.
>
>
> [1] Wang, Y., & Zhang, M. An empirical study of realized GNN expressiveness. ICML 2024.
>
> [2] Horn, Max, et al. "Topological graph neural networks."  ICLR 2022.
>
> [3] Zhang, Bohang, et al. "Rethinking the expressive power of GNNs via graph biconnectivity." ICLR 2023.

---

### Note · Authors · 2025-08-14

Dear Area Chair and Reviewers,

We sincerely appreciate this opportunity to provide final remarks and thank all reviewers for their valuable feedback.

**Contribution & Recognition**: Through comprehensive experiments and systematic analysis, we demonstrate that vision models possess remarkable yet underutilized capabilities for graph structural understanding. Reviewer ofCS recognized this as "very foundational work, which is well-justified, rigorously presented, and methodically executed". Reviewer 3Uke praised our "clear and original contribution" with "comprehensive empirical results." Reviewer Aeju acknowledged the work provides "compelling insights into the future direction of graph AI." We provided detailed responses to all questions and concerns raised during the review, including additional experiments and clarifications that we believe address the reviewers' thoughtful feedback.

**Additional Validation**: During the discussion period, we conducted extensive experiments to address reviewer concerns, including scaling studies up to 212M parameters and multi-resolution analysis.  We demonstrated that vision models consistently outperform GNNs on tasks requiring global structural awareness. Across all four tasks, vision models maintain strong performance even on graphs 3-5× larger than training, while GNNs experience significant degradation. This performance gap persists regardless of model capacity (212M for GNNs vs 25.6M for ResNet), confirming that architectural approaches rather than size drive these fundamental differences.

**Broader Impact**: Our work provides fundamental insights into how different computational paradigms access graph structure. As Reviewer ofCS noted, our "discussion on vision models and layout algorithms is insightful, and paves the way for further research." By establishing that visual processing offers a complementary path to graph understanding, our findings could spark interest across vision, graph learning, and graph visualization communities. These insights point toward promising directions including graph-specific visual architectures and theoretical foundations connecting geometric properties of layouts to their learnability.

We are grateful for the constructive feedback from all reviewers, which has significantly improved our work. We remain committed to incorporating all reviewer suggestions in our revision.

Thank you for your time and consideration.

Best regards,

The Authors

---

### Decision · Program_Chairs · 2025-09-17

**Decision:**

Accept (poster)

**Comment:**

This paper explores the potential of vision models for graph structural understanding, introducing the GraphAbstract benchmark to evaluate global topological reasoning across tasks like symmetry detection and bridge counting. Reviewers found the work original and rigorous, with ofCS highlighting its foundational contributions and 3Uke and Aeju appreciating the benchmark's conceptual clarity. Initial concerns about computational fairness and layout-induced confounds were addressed through thorough rebuttals and experiments, including scaling studies and resolution analysis. While limitations remain in formalizing the role of layout algorithms, the paper convincingly demonstrates that vision models offer complementary strengths to GNNs. Given its empirical depth and conceptual novelty, this paper is recommended for poster presentation, and future work should pursue theoretical grounding and hybrid modeling strategies.